# Towards Combating Frequency Simplicity-biased Learning for Domain Generalization

**Xilin He[1], Jingyu Hu[1], Qinliang Lin[1], Cheng Luo[1], Weicheng Xie[1,2,3†]**
**Siyang Song[4], Muhammad Haris Khan[5], Linlin Shen[1,2,3]**

Computer Vision Institute, School of Computer Science & Software Engineering, Shenzhen University[1]
Shenzhen Institute of Artificial Intelligence and Robotics for Society[2]
Guangdong Provincial Key Laboratory of Intelligent Information Processing[3]
University of Exeter[4], Mohamed bin Zayed University of Artificial Intelligence[5]
wcxie@szu.edu.cn

## Abstract

Domain generalization methods aim to learn transferable knowledge from source domains that can generalize well to unseen target domains. Recent studies show that neural networks frequently suffer from a simplicity-biased learning behavior which leads to over-reliance on specific frequency sets, namely as frequency shortcuts, instead of semantic information, resulting in poor generalization performance. Despite previous data augmentation techniques successfully enhancing generalization performances, they intend to apply more frequency shortcuts, thereby causing hallucinations of generalization improvement. In this paper, we aim to prevent such learning behavior of applying frequency shortcuts from a data-driven perspective. Given the theoretical justification of models' biased learning behavior on different spatial frequency components, which is based on the dataset frequency properties, we argue that the learning behavior on various frequency components could be manipulated by changing the dataset statistical structure in the Fourier domain. Intuitively, as frequency shortcuts are hidden in the dominant and highly dependent frequencies of dataset structure, dynamically perturbating the over-reliance frequency components could prevent the application of frequency shortcuts. To this end, we propose two effective data augmentation modules designed to collaboratively and adaptively adjust the frequency characteristic of the dataset, aiming to dynamically influence the learning behavior of the model and ultimately serving as a strategy to mitigate shortcut learning. Code is available at [AdvFrequency](AdvFrequency).

## 1 Introduction

Previous studies have shown that the performance of deep neural networks usually degrades when processing previously unseen data whose distribution differs radically from the training data distribution, which largely constrains the deployment of well-trained deep models [11, 10]. Domain generalization (DG) aims to learn models with transferable knowledge, utilizing multiple source domains, that can generalize well to unseen test data to address this problem [19, 8]. In this paper, we focus on a more challenging scenario where only one source domain is available for training, namely as single source domain generalization (SDG).

---

†Corresponding author

38th Conference on Neural Information Processing Systems (NeurIPS 2024).

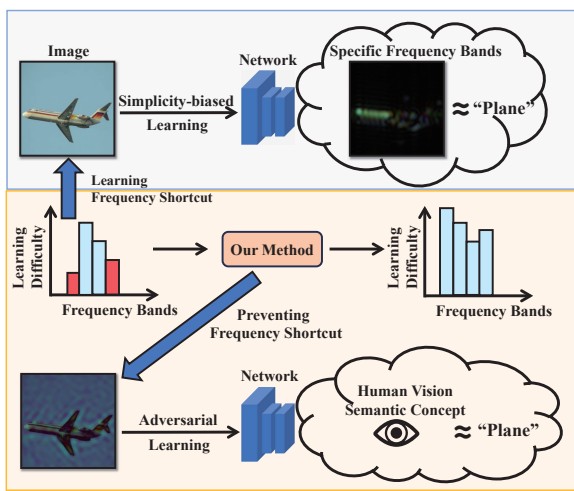

Figure 1: Illustration of model's frequency simplicity-biased learning and motivation of our method. Deep models tend to learn the simplest solution (e.g. the specific class-wise most distinctive frequency bands) in classification tasks instead of the semantic cues. Our method proposes to adaptively modify the learning difficulty of different frequency components to prevent frequency shortcut learning.

The domain generalization problem can be conventionally solved by learning domain invariant features [23, 7], conducting adversarial training [20, 15], employing meta-learning [4] or data augmentation [37, 16]. Among the proposed techniques, data augmentation is prone to be the simplest and the most effective one. Specifically, [12] proposes to mix various augmented version of images to enhance robustness. [16] proposes to generate diverse stylized samples to improve the generalization ability of neural networks. Recently, an interest in understanding the learning dynamic of neural networks to improve generalization has arisen [40, 26, 35, 31]. Neural networks are found to fit a function from low frequencies first in the regression task, which is widely known as F-principle [40] or spectral bias [26]. [28] reveal that neural networks tend to learn simple but effective patterns, such as the shortcut solutions that disregard semantics related to the problem at hand but are simpler to minimize the optimization loss, which leads to over-fitting and generalization performance degradation. With the generalization performance improvement brought by previous data augmentation techniques [12, 16], it's widely believed that previous successful data augmentation techniques can effectively prevent shortcut learning. However, a recent study on frequency shortcuts [35], which is the certain set of frequency bands used specifically to classify certain classes in classification task, empirically disproves this assumption. They have demonstrated that previous data augmentation techniques [12, 16], despite gaining generalization performance improvement, learn more frequency shortcuts instead and cause the hallucination of generalization improvement.

To this end, we aim to develop data augmentation techniques capable of preventing the learning of the frequency shortcuts, which are neglected by previous works and thereby achieving enhanced generalization performances. Since [24] has introduced a mathematical formulation demonstrating that the learning dynamics of neural networks are biased towards dominant frequencies of the statistical structure of the dataset, which maybe not necessarily low frequencies, the study [24] offers a theoretical basis for the manipulation of a model's learning behavior across different frequency components. In this paper, building upon the theoretical underpinning provided by [24], we attempt to modify the frequency spectrum of the dataset statistical structure with aggressive frequency data augmentation, as this would affect the dominant frequencies of the dataset and thereby change the model's learning strength on different frequencies. By dynamically modifying the frequency property of the dataset, we would be able to adaptively manipulate model's learning behavior on various frequency components and thereby prevent the application of frequency shortcuts by suppressing the learning on the highly dependent frequencies. To this end, we propose two effective and practical adversarial frequency augmentation modules, i.e., Adversarial Amplitude Uncertainty Augmentation (AAUA) and Adversarial Amplitude Dropout (AAD). These modules are designed to effectively and aggressively alter the frequency characteristic, aiming to dynamically manipulate the model's

learning behavior on over-reliance frequency components and ultimately help prevent the learning of frequency shortcuts. Our core contributions are summarized as follows:

- Given the theoretical justification of neural network's learning behavior on frequency short-cuts, we propose two effective data augmentation modules, adversarial amplitude uncertainty augmentation module (AAUA) and adversarial amplitude dropout module (AAD) to dynamically and implicitly regularize and suppress the model's over-reliance to specific frequency bands, serving as the pioneering work to combat frequency shortcuts for domain generalization.

- Our proposed method is evaluated under a variety of DG tasks and datasets, ranging from image classification to instance retrieval. The results show the superiority of the proposed augmentation algorithm over state of the arts (SOTAs).

## 2   Related Work

**Domain Generalization:** Domain generalization (DG) aims to learn transferable knowledge that can generalize well to unseen target domains. In general, the DG problem has two settings: multi-source domain generalization and single domain generalization (SDG), distinguished by the number of available source domains during training. The SDG is considered as a more realistic scenario with greater difficulty due to the lack of diversity in data. Inspired by domain adaptation methods, early studies attempt to carry out domain alignment to learn domain invariant features by reducing the bias of feature distributions among multiple source domains [20, 15, 21] with adversarial training [15, 29] or minimizing distance metrics between distributions [20, 21]. In addition, self-supervised learning [1, 17] and meta-learning [4, 30] have also been studied.

Recently, data augmentation [50, 37, 51], especially style augmentation [37, 51, 22, 16], has been widely studied and shows appealing performances. L2D [37] diversifies image styles by playing a min-max game with mutual information. DSU [22] proposes to perform style augmentation in the feature level and augment feature statistics with uncertainty estimation to simulate the domain shifts. Rising methods resort to data augmentation in the frequency domain [41, 14, 38, 34, 43]. Represented by FACT [38], the vast majority of the frequency augmentation methods resort to randomly swapping or mixing the amplitude spectrum maps within the same batch. SADA [43] proposes to inject adversarial perturbations into the amplitude spectrum maps, intending to suppress frequency sensitivity to gain generalization performance improvement. However, previous successful data augmentation techniques cannot successfully prevent the learning and application of frequency shortcuts despite gaining generalization performance improvement. Wang et. al. [35] demonstrate that data augmentation techniques such as AugMix [13] and style augmentation [37] surprisingly learn more frequency shortcuts and cause hallucinations of generalization ability improvement.

**Shortcut Learning:** Shortcuts in classification tasks are known as the spurious correlations between the input data and ground truth labels, rather than the semantic information [5]. Since models that apply shortcuts tend to give predictions based on the presence of the specific shortcuts instead of the transferable semantic content of data, shortcut learning could damage the generalization ability of learned models. Recent studies look into the frequency domain to understand the shortcut learning behavior. [38] prove that neural networks tend to fit a function from low-frequencies priorly and then higher frequencies in the regression tasks, which is known as the F-principle. [24] mathematically formulated the finding that the neural network's learning behavior is biased towards the dominant frequencies of the dataset and could be manipulated. [35] demonstrate that models' learning order of different frequency components in the classification tasks is data-driven and models tend to learn the simplest and the most class-wise distinctive frequency components to give predictions, which is known as the frequency shortcut learning. They further reveal that previous successful data augmentation techniques cannot prevent the learning and application of frequency shortcuts.

## 3   Analysis

In this section, we provide clarification on the definition of the discussed 'frequency' in this paper and a theoretical analysis of network's learning behavior on frequency components following [24].

## 3.1 Clarification of the Frequency in this Work

**Clarifying the definition of frequency.** It is worth noting that wide misconceptions between spatial frequency and responsive frequency exist in the research community. We would have to emphasize that the frequency discussed in this paper is not the same as the frequency discussed in the well-known F-principle [40] and spectral bias [26]. The frequency discussed in this paper is the nature of the image data itself, known as **spatial frequency**, which is the characteristic of the image data processed by Fourier transformation. In contrast, the term "frequency" in F-principle [40] and spectral bias [26] refers to the concept of **responsive frequency or function frequency**. This term describes the characteristics of the mapping function learned by networks, which transforms input images into probability vectors. It signifies the smoothness of the learned mapping by the trained networks.

**Learning behavior on frequencies.** Since the two frequency definitions are different, the claim that networks prefer to learn low frequencies first [40] does not necessarily hold true. The claim holds true for responsive frequency. As for spatial frequency, frequency components that the model prioritizes to learn are not necessarily low-frequency components but are determined by the dataset's properties [24]. However, in real-world datasets, as low-frequency components are often the main contributors to the Fourier spectrum of the singular vectors to a dataset's statistical structure, neural networks might prioritize learning low-frequency components of images in most cases.

**Tasks of concern.** Previous works on the responsive frequency focus on the regression task. In contrast, this paper aligns with recent studies on spatial frequency [35], which mainly focus on the classification task.

## 3.2 Theoretical Justification

In this section, we refer to the theoretical observation from [24] as the justification for the provability of manipulating the model's learning behavior on various frequency components and preventing frequency shortcut learning.

**Preliminaries.** For the sake of clarity, we first provide descriptions of basic variables.

*(i)* Two-dimensional images are vectorized to $1 \times n^2$.

*(ii)* To represent the statistical structure of a dataset with $p$ classes, following Pinson et. al. [24], a $p \times n^2$ input-output correlation matrix $\Sigma^{yx}$ is adopted. Given one-hot encoded ground truth labels $y$, each row of $\Sigma^{yx}$ contains the average of images in the corresponding class. The input-input correlation matrix $\Sigma^{xx}$ and prediction-input correlation matrix $\Sigma^{\hat{y}x}$ is similarly defined.

*(iii)* In an SVD basis, $\Sigma^{yx}$ can be decomposed as $\Sigma^{\hat{y}x} = USV^T$, where $S$ and $V$ contain the singular values and right singular vectors, respectively. The first $p$ rows of $V^T$, denoted as $\phi^\alpha$, $\alpha \in 0, ..., p-1$, are the principal components of the dataset structure. In the SVD basis of $\Sigma^{\hat{y}x}$, the prediction-output matrix $\Sigma^{\hat{y}x}$ can be decomposed in a way similar to SVD with $U$ and $V^T$ as $\Sigma^{\hat{y}x} = UAV^T$. $s_\alpha$ and $a_\alpha$ denote the $\alpha$-th diagonal elements in $S$ and $A$, respectively.

**Data-driven frequency learning behavior.** For the learning dynamics of networks, the learning behavior is biased toward the frequencies that are dominant in the Fourier spectrum of the singular vectors derived from the statistical structure of a dataset. This is mathematically formulated as [24]:

$$\frac{d\,|(Q\,r)_j|^2}{dt} = c|(Q\,r)_j|^2\,|(Q\,\phi^\alpha)_j|(s_\alpha - a_\alpha), \tag{1}$$

where $Q$ and $r$ denote the vectorized 2D discrete Fourier transformation and the convolution kernel in a network. $j$ and $c$ denote the indices of the frequencies contributing to $\phi^\alpha$ and an irrelevant constant representation for simplicity, respectively. In Eq. 1, $s_\alpha$ can be considered as the input, while $(Q\,r)_j$ represents the Fourier coefficients of the kernel $r$, indicating the kernel's reliance on the $j$-th frequency. As analyzed by Pinson et al. [24], when the convolution kernels are initialized with small values, $a_\alpha$ and $(Q\,r)_j$ would be small. In this case, the learning dynamic is mainly driven by the frequency property of the dataset structure $(Q\,\phi^\alpha)_j$. Thereby, the model would be highly dependent on the dominant frequencies that contribute most to the singular vectors $\phi$ of the dataset.

**Risk of applying frequency shortcuts.** Intuitively, in the natural image datasets, strong correlations between patterns could exist. These kinds of correlations could exhibit themselves in the statistical structure of the datasets, and thereby be picked up by the networks during the learning process [5].

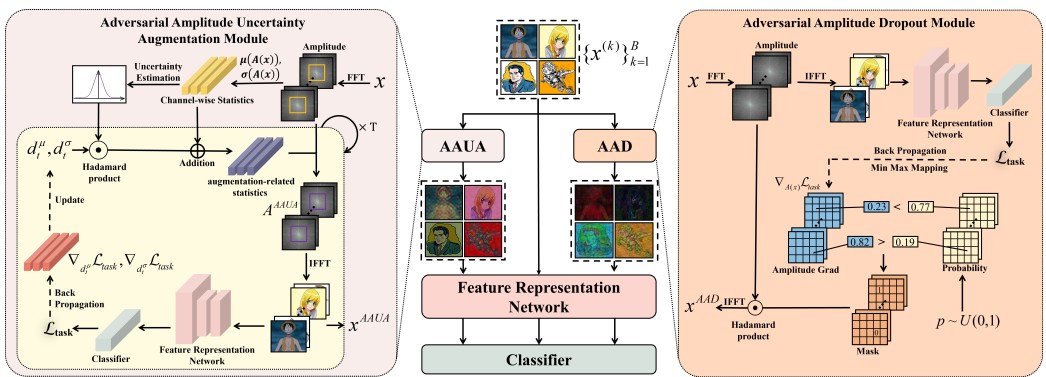

Figure 2: Overview of the proposed adversarial frequency augmentation modules.

Given the formulated biased learning behavior in Eq. 1, the focus of learning is biased toward frequency components that are dominant in the singular vectors of the Fourier spectrum of the dataset. As it is acknowledged that specific frequency components beneficial for the classification task in specific domains [5, 35] are outstanding in the dataset structure, a model directly trained on these images might take the risks of being over-reliance on these frequency components.

Recently, [35] has highlighted a tendency in networks to leverage shortcuts of frequencies that are specific and highly discriminative frequency components across classes, rather than semantic cues. This frequency shortcut learning phenomenon is evidenced in Eq. 1, which shows that the learning process prioritizes frequency components specific to the singular vectors of $\Sigma^{yx}$ with higher singular values [24]. Consequently, the model captures the most discriminative frequency components in the spectrum of the dataset structure, potentially leading to the adoption of frequency shortcuts.

**Leveraging the learning behavior as a shield against frequency shortcuts.** As the learning behavior in Eq. 1 is mathematically defined, it would be difficult to go against it with a fixed dataset structure. However, if the model is deemed to capture frequency shortcuts hidden in the dominant frequencies on a static dataset, natural questions emerge: *Can we adaptively modify the dataset to reduce the possibility of frequency shortcuts by leveraging a biased frequency learning rule? Can we construct a dynamic dataset to modify its frequency property in each learning iteration?*

**Motivation in preventing frequency shortcuts.** To this end, this paper attempts to combat the frequency shortcut learning by dynamically modifying the frequency characteristic of the dataset structure. As shown in Eq. 1, the gradient is dynamically influenced by the highly coupled interplay of dataset frequency characteristics and model weight. Given the dynamic and decoupled learning process in Eq. 1, we attempt to make use of adversarial learning to bridge between model weight and the dataset frequency characteristics.

To effectively remould the spectrum map of the dataset structure, we propose to conduct aggressive augmentation on the frequency spectrum, instead of simply injecting noises. Further, different from prior frequency augmentation methods [43], this paper utilizes adversarial gradients w.r.t. the amplitude spectrum maps to locate the over-reliance frequency components, instead of following simple assumptions to mask specific frequency bands. However, it could be technically difficult to directly manipulate the frequency characteristic of the whole dataset statistical structure due to computational resource constraints when backpropagating on the entire dataset. Therefore, we propose two specific in-batch adversarial frequency augmentation modules to effectively modify the dataset frequency characteristic.

## 4 Methodology

The main goal in single source domain generalization is to train a model $f$ on a source domain, which should generalize well in various previously unseen target domains. We denote the source domain as $D_S = \{(x, y)\}$, $x \in \mathbb{R}^{W \times H \times C}$ is a source image, $W$, $H$ and $C$ are the width, height and the number of channels of source images and $y$ is the corresponding label.

For convenience, we use $\mathcal{F}(\cdot)$ to denote Fast Fourier Transformation (FFT). $A(\cdot)$ and $P(\cdot)$ denote the calculation of amplitude and phase, respectively. In the following sections, we propose the module

of Adversarial Amplitude Uncertainty Augmentation (AAUA) and its complementary module of Adversarial Amplitude Dropout (AAD).

## 4.1 Adversarial Amplitude Uncertainty Augmentation (AAUA)

As low-frequency components are the main contributors in natural image datasets, to effectively modify the frequency spectrum of dataset structure, we propose to inject adversarial noises into the low-frequency components. Instead of directly injecting noises [43] into the frequency spectrum map, we propose to combine adversarial learning with instance normalization to craft aggressive augmentations to effectively disrupt the overall dataset structure and thereby prevent the learning of frequency shortcuts hidden in the low-frequency components.

Different from previous frequency augmentation methods [43, 34] that treat each amplitude spectrum map as deterministic values, we hypothesize that statistics of each low-frequency part of amplitude spectrum maps are samples drawn from a Gaussian distribution, based on which we adversarially draw new statistics to reconstruct the augmented low-frequency spectrum maps to properly model the uncertainty of domain shifts rich in diversity and hardness.

**Adversarial low-frequency component transformation.** Given a batch of input images $\{x^{(k)}\}_{k=1}^B$, where $B$ denotes the batch size, we first obtain the amplitude spectrum maps. We employ $M_{\text{low}}$ and $M_{\text{high}} = 1 - M_{\text{low}}$ as the masks to separate low-frequency and high-frequency components as:

$$M_{\text{low}}(i,j) = \begin{cases} 1 & \text{if } i \in [h_{min}, h_{max}], j \in [w_{\min}, w_{\max}] \\ 0 & \text{otherwise} \end{cases} \tag{2}$$

where $h_{\min} = \lfloor (1-\lambda)H/2 \rfloor$, $h_{\max} = \lfloor (1+\lambda)H/2 \rfloor$, $w_{\min} = \lfloor (1-\lambda)W/2 \rfloor$ and $w_{\max} = \lfloor (1+\lambda)W/2 \rfloor$ denote the lower and upper bounds to obtain height and width of low-frequency mask. $\lfloor \rfloor$ denotes the round-down operator. $\lambda$ controls the size of the mask. As it would be difficult to precisely determine which band is low-frequency or not, $\lambda$ is drawn from a uniform distribution $U(0, 0.5)$. Then, we calculate the channel-wise statistics of the low-frequency part following:

$$\begin{cases} \mu(A(x)) = \dfrac{1}{\lambda^2 HW} \displaystyle\sum_{i=h_{min}}^{h_{max}} \sum_{j=w_{min}}^{w_{max}} A(x)(i,j,:) \\[2ex] \sigma(A(x)) = \dfrac{1}{\lambda^2 HW} \displaystyle\sum_{i=h_{min}}^{h_{max}} \sum_{j=w_{min}}^{w_{max}} [A(x)(i,j,:) - \mu(A(x))]^2 \end{cases} \tag{3}$$

where $\mu$ and $\sigma$ denote mean and variance, respectively. We further use the standard deviations of these channel-wise statistics for the uncertainty estimation $\sum_\mu \in \mathbb{R}^C$ and $\sum_\sigma \in \mathbb{R}^C$:

$$\begin{cases} \sum_\mu^2(A(x)) = \dfrac{1}{B} \displaystyle\sum_{k=1}^B [\mu(A(x^{(k)})) - \mathbb{E}_b(\mu(A(x)))]^2 \\[2ex] \sum_\sigma^2(A(x)) = \dfrac{1}{B} \displaystyle\sum_{k=1}^B [\sigma(A(x^{(k)})) - \mathbb{E}_b(\sigma(A(x)))]^2 \end{cases} \tag{4}$$

where $x^{(k)}$ and $\mathbb{E}_b(\cdot)$ denote the $k$-th image in a batch and the expectation operator within this batch. Then, augmentation-related statistics are updated following:

$$\begin{cases} \mu'(A(x)) = \mu(A(x)) + d_t^\mu \odot \sum_\mu \\ \sigma'(A(x)) = \sigma(A(x)) + d_t^\sigma \odot \sum_\sigma \end{cases} \tag{5}$$

where $d_t^\mu$ and $d_t^\sigma$ control the intensity and direction of the adversarial perturbations to the channel-wise mean and variance at the $t$-th iteration of AAUA, which are to be optimized. $\odot$ denotes the Hadamard product. Then the augmented amplitude is derived as:

$$\begin{cases} A_{\text{low}}^{\text{AAUA}} = \left( \dfrac{A_{\text{low}}(x) - \mu(A(x))}{\sigma(A(x))} \right) \sigma'(A(x)) + \mu'(A(x)) \\[2ex] A^{\text{AAUA}} = M_{\text{low}} \odot A_{\text{low}}^{\text{AAUA}} + M_{\text{high}} \odot A(x) \end{cases} \tag{6}$$

where $A_{\text{low}}(x) = A(x) \odot M_{\text{low}}$ denotes the low frequency parts of $A(x)$. We can formulate the $t+1$-th optimization step of adversarial channel-wise statistics control factors $\{d_{t+1}^\mu, d_{t+1}^\sigma\}$ as:

$$\begin{cases} d_{t+1}^\mu = d_t^\mu + sign(\nabla_{d_t^\mu} \mathcal{L}_{\text{task}}(x^{\text{AAUA}}, y; f)) \\ d_{t+1}^\sigma = d_t^\sigma + sign(\nabla_{d_t^\sigma} \mathcal{L}_{\text{task}}(x^{\text{AAUA}}, y; f)) \end{cases} \tag{7}$$

where $x^{\text{AAUA}} = \mathcal{F}^{-1}(A^{\text{AAUA}}, P(x))$ is generated with augmented amplitude $A^{\text{AAUA}}$ and original phase $P(x)$ by Inverse Fast Fourier Transformation (IFFT) $\mathcal{F}^{-1}$, and $sign()$, $\nabla$ and $\mathcal{L}_{\text{task}}$ denote the sign function, gradient operator and the task-related loss for adversarial gradient calculation, e.g. cross-entropy loss for image classification. We use the above optimization for $T$ steps to generate the AAUA-augmented image $x^{\text{AAUA}}$.

### 4.2 Adversarial Amplitude Dropout (AAD)

Despite AAUA shifting the network's attention away from the low-frequency components, it may lead to models' over-fitting into the imperceptible noises hidden in high-frequency bands, leading to a decline in models' generalization abilities. Therefore, to prevent the learning of frequency shortcuts in the high-frequency components, we further propose an Adversarial Amplitude Dropout (AAD) module to adaptively modify the frequency spectrum maps with backward gradients to estimate the model's frequency characteristics, which allow us to dropout the highly dependent frequency components to remove the learned frequency shortcuts.

Different from prior frequency masking methods [14, 41] that simply mask specific bands, we attempt to adaptively mask the over-reliance frequency bands. To locate the critical frequency bands, inspired by [43, 2], we utilize the gradients w.r.t. amplitude as the surrogate of model's frequency sensitivity map across frequency bands:

$$g_a = \nabla_{A(x)} \mathcal{L}_{\text{task}}(x, y; f) \tag{8}$$

Since the larger magnitude of these gradients represents a stronger sensitivity to the corresponding frequency band, a larger probability of this frequency band is preferred to be dropout. Further, to craft more augmented samples with various learning difficulties, a random threshold for dropping frequency bands is introduced. Thereby, we generate a dropout mask with the amplitude gradients $g_a$ and a random threshold $p$ drawn from a uniform distribution $U(0, 1)$ following:

$$M^{\text{AAD}}(i, j) = \begin{cases} 1 & \text{if } s(g_a(i,j)) < p(i,j) \\ 0 & \text{otherwise} \end{cases} \tag{9}$$

where $M^{\text{AAD}}(i, j)$ denotes the mask value at the coordinate of $[i, j]$ and min max mapping $s(x) = (x - \max(x))/(\max(x) - \min(x))$. Then the dropout amplitude is derived by Hadamard product:

$$A^{\text{AAD}}(x) = A(x) \odot M^{\text{AAD}} \tag{10}$$

where $A^{\text{AAD}}(x)$ denotes the dropout amplitude of image $x$. We further obtain the augmented sample $x^{\text{AAD}}$ with dropout amplitude and the original phase through IFFT following:

$$x^{\text{AAD}} = \mathcal{F}^{-1}(A^{\text{AAD}}(x), P(x)) \tag{11}$$

where $\mathcal{F}^{-1}$ denotes the IFFT. For clarity of narrative, the pseudo-code of generating augmentation samples and samples generated by AAUA and AAD are shown in the supplementary material.

### 4.3 Model Training

To further regularize the prediction consistency between benign and augmented images, a Jensen-Shannon divergence consistency loss $\mathcal{L}_{JS}$ [13] is also adopted. Therefore, we can formulate the total training loss as:

$$\mathcal{L} = \mathcal{L}_{\text{task}} + \mathcal{L}_{\text{JS}} \tag{12}$$

where $\mathcal{L}_{\text{task}}$ denotes the task-related loss, e.g. cross-entropy loss for image classification and instance retrieval. Both of the proposed modules generate augmented images for training in each iteration.

## 5 Experiments

We first conduct cross-domain image classification and instance retrieval experiments to study the proposed method's performance. We compare our method with traditional data augmentation based methods [13, 33, 39], image-level style augmentation based methods [25, 37], feature-level style augmentation methods [45, 22, 44], two most relative frequency augmentation-based methods [38, 43] and other representative methods [23, 1]. Then we evaluate the frequency shortcuts following Wang's metric [35]. We further provide insights into models' data-driven frequency characteristics with the proposed AAUA and AAD. We also carry out detailed ablation studies.

Table 1: Experimental results on Digits, PACS and CIFAR-10-C.

| Method | Digits | PACS | CIFAR-10-C |
|---|---|---|---|
| ERM | 49.29 | 46.09 | 54.08 |
| GUD[33]$_{\text{NeurIPS'18}}$ | 55.67 | 57.56 | 59.91 |
| M-ADA[25]$_{\text{CVPR'20}}$ | 59.49 | 59.40 | 64.65 |
| AugMix[12]$_{\text{ICLR'20}}$ | 72.45 | 62.47 | 75.28 |
| RandConv[39]$_{\text{ICLR'21}}$ | 72.88 | 62.76 | 71.23 |
| L2D[37]$_{\text{CVPR'21}}$ | 74.45 | 64.74 | 72.88 |
| FACT[38]$_{\text{CVPR'21}}$ | - | 59.10 | - |
| DSU[22]$_{\text{ICLR'22}}$ | - | 53.70 | - |
| SADA[43]$_{\text{AAAI'23}}$ | 76.56 | 65.71 | 77.33 |
| FFM[36]$_{\text{WACV'23}}$ | 73.45 | 70.4 | 77.77 |
| ALT[6]$_{\text{WACV'23}}$ | 72.49 | 64.72 | - |
| ABA[3]$_{\text{ICCV'23}}$ | 74.76 | 66.02 | - |
| AdvST[46]$_{\text{AAAI'24}}$ | 80.10 | 64.10 | - |
| Ours | **81.39** | **67.91** | **78.33** |

Table 2: Experimental results of the task of person re-ID on Market1501 and DukeMTMC. We follow the benchmark by [44].

| Method | Market→Duke | | | | Duke→Market | | | |
|---|---|---|---|---|---|---|---|---|
| | mAP | R1 | R5 | R10 | mAP | R1 | R5 | R10 |
| OSNet | 27.9 | 48.2 | 62.3 | 68.0 | 25.0 | 52.8 | 70.5 | 77.5 |
| + RandomErase | 20.5 | 36.2 | 52.3 | 59.3 | 22.4 | 49.1 | 66.1 | 73.0 |
| + DropBlock | 23.1 | 41.5 | 56.5 | 62.5 | 21.7 | 48.2 | 65.4 | 71.3 |
| + MixStyle | 28.0 | 49.5 | 63.6 | 68.8 | 27.5 | 57.4 | 74.1 | 80.1 |
| +EFDMix | 29.9 | 50.8 | 65.0 | 70.3 | 29.3 | 59.5 | 76.5 | 82.5 |
| +StyleNeophile | 29.7 | 50.6 | 65.4 | 74.2 | 32.2 | 64.7 | 80.2 | 89.1 |
| +DSU | 31.0 | 53.0 | 66.6 | 71.7 | 29.3 | 59.9 | 77.2 | 82.8 |
| +AdvStyle | 33.2 | 55.7 | 69.1 | 73.5 | 32.0 | 63.2 | 79.7 | 85.0 |
| +SADA | 33.1 | 53.2 | 67.4 | 72.5 | 30.0 | 62.7 | 78.4 | 84.2 |
| +Ours | **33.8** | **56.3** | **70.1** | **75.8** | **34.2** | **64.9** | **81.8** | **90.1** |

## 5.1 Datasets and Implementation Details

**Datasets for image classification and instance retrieval:** We evaluate the performance of the proposed method on three cross-domain image classification benchmarks of PACS [19], Digits and CIFAR-10-C [10]. We conduct the cross-domain instance retrieval task on person re-identification (re-ID) datasets of Market1501 [47] and DukeMTMC [27] with OSNet [48].

**Implementation Details** In all the experiments, we set the iterative optimization steps $T$ of AAUA as 5. For a fair comparison, the number of augmented samples used in each iteration is set as 3, following [43]. We adopt the same network architectures in previous works [43, 44]. Please see the supplementary material for more implementation details.

## 5.2 Comparisons with State-of-The-Arts

We conduct experiments of cross-domain image classification and person re-ID in five benchmarks. models trained with our generated data obtain significant generalization performance improvement (Tab. 1). Specifically, our method outperforms SADA [43] and FACT [38] with large margins of 2.20% and 8.81% in PACS. Following [45, 44], we conduct cross-domain instance retrieval of person re-ID task on Market1501 [47] and DukeMTMC [27]. Experiment results are shown in Tab. 2. Our method outperforms previous SOTA StyleNeophile [16] with a large margin, boosting the mAP on Market→Duke from 29.7 to 33.8, further validating the effectiveness of the proposed method.

## 5.3 Frequency Shortcut Evalutation

Wang et al. [35] propose to use TPR and FPR on the dominant frequency maps (DFM) filtered test sets of ImageNet-10 to evaluate how many frequency shortcuts the classifier is induced to learn and apply. Semantic information of DFM-filtered samples is severely damaged. In this paper, we use the average of class-wise TPR and FPR on the DFM-filtered test sets of ImageNet-10 to quantify how many frequency shortcuts are applied. Higher TPR and FPR values indicate that more frequency shortcuts are applied. Samples from the DFM-filtered test set are shown in the supplementary material.

The frequency shortcuts quantitative results of the standard network of ResNet-18, AugMix, style augmentation and ours are shown in Tab. 3. As shown in Tab. 3, the combination of AAUA and AAD effectively suppresses the learning of frequency shortcuts, while previously successful data augmentation techniques apply more frequency shortcuts, causing the hallucinations of generalization ability improvement. It's worth noting that when AAUA or AAD is solely applied, the model achieves more frequency shortcuts. As AAUA solely perturbates the low-frequency components, leaving high-frequency components unchanged. Hence, using AAUA alone, without AAD, may lead to exploring frequency shortcuts in high-frequency bands.

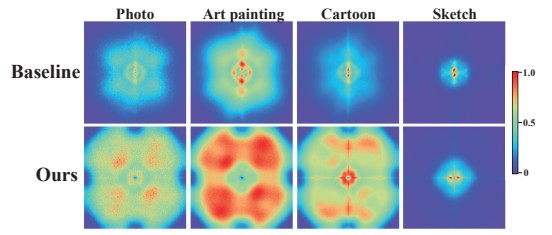
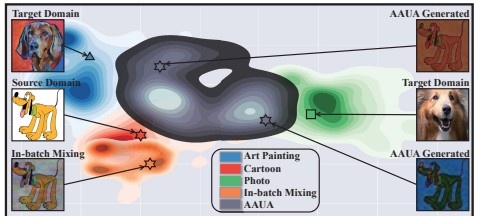

(a) Frequency sensitivity maps      (b) Feature manifolds of augmented samples

Figure 3: (a) Visualizations of model frequency sensitivity maps on source (Photo) and target domains (Art painting, Cartoon, Sketch). (b) Features manifolds of augmented samples and various domains.

## 5.4 Empirical Analysis

### 5.4.1 Frequency sensitivity of trained models

To validate that modifying the frequency property of the dataset structure could indeed affect the model's preferences on various frequency components, we further show the frequency sensitivity of baseline models and models trained with our method. Following Zhang et al. [2], we obtain frequency sensitivity maps with backward gradients w.r.t amplitude spectrum maps as surrogates of model's frequency sensitivity. It's worth noting that the frequency sensitivity maps are derived after a softmax function. Therefore, it would only show relative sensitivity across frequency bands, but not indicate the application of frequency shortcuts.

### 5.4.2 Augmentation strength of different augmentation methods

To validate the effectiveness of modifying the dataset frequency property with AAUA, we further compare the hardness between previous in-batch mixing frequency augmentation methods (e.g. FACT [38]) and our proposed AAUA. As shown in Fig. 3(b), we use one sample in the source domain to generate sufficient augmented samples. Compared with in-batch mixing methods, feature embeddings of samples generated with the proposed AAUA span across different domains and are further from the source domain, indicating that the proposed AAUA generates augmented samples with greater hardness and diversity.

Table 3: Frequency shortcuts quantification on the DFM-filtered test set of ImageNet-10.

| Method | Avg TPR ↓ | Avg FPR ↓ |
|---|---|---|
| ResNet-18 | 0.286 | 0.041 |
| ResNet-18+AugMix | 0.242 | 0.046 |
| ResNet-18+Style Augmentation | 0.250 | 0.090 |
| ResNet-18+FACT | 0.274 | 0.048 |
| ResNet-18+AAUA | 0.265 | 0.068 |
| ResNet-18+AAD | 0.280 | 0.083 |
| ResNet-18+Ours | **0.212** | **0.030** |

Table 4: Ablation studies conducted on PACS dataset. PD denotes performance degradation compared with the full method.

| AAD | AAUA | $\mathcal{L}_{JS}$ | PT | AP | CT | SC | Avg.(PD) |
|---|---|---|---|---|---|---|---|
| ✔ | | | 47.38 | 72.60 | 78.28 | 54.97 | 63.31 (-4.60) |
| | ✔ | | 53.91 | 72.92 | **79.35** | 55.92 | 65.53 (-2.38) |
| ✔ | ✔ | | 53.73 | 74.59 | 78.30 | 58.78 | 66.35 (-1.56) |
| ✔ | | ✔ | 51.75 | 71.40 | 75.57 | 56.91 | 63.91 (-4.00) |
| | ✔ | ✔ | 57.83 | 74.85 | 78.44 | 54.75 | 66.47 (-1.44) |
| ✔ | ✔ | ✔ | **58.41** | **75.71** | 78.59 | **58.92** | **67.91** |

## 5.5 Ablation Study

We conduct ablation studies in Tab. 4 on the proposed adversarial frequency augmentation modules to study the performance of each component in the proposed method. As shown in Tab. 4, the generalization performance on PACS dataset degrades by $1.44\%$ or $4.00\%$ without the proposed AAD or AAUA module, validating the effectiveness of the proposed augmentation modules. Analysis of the iterative optimization steps $T$ in AAUA and the proportion of augmentation modules' impact on the generalization performance is shown in the supplementary material.

## 6 Conclusion

In this paper, we attempt to counteract models' simplicity-biased learning behavior of applying frequency shortcuts from the perspective of data-driven and learning difficulty, with previous successful

data augmentation techniques failing to do so. Since neural networks take up the simplest and the most class-wise distinctive frequency patterns as frequency shortcuts, we propose to adaptively modify the learning difficulty of different frequency components by combining adversarial training with instance normalization (AAUA) and dropout operation (AAD) in the frequency domain. By effectively preventing the simplicity-biased learning of frequency shortcuts and manipulating the trained models' frequency characteristics, the experimental results on five public benchmarking datasets ranging from classification to instance retrieval demonstrate the superiority of our method.

**Limitations and future work.** Despite the proposed method indirectly preventing frequency shortcuts from a data-driven perspective, the direct location of frequency shortcuts remains a challenging problem with no effective solutions in the field. In the future, we would like to investigate more essential properties of frequency shortcuts to help locate them effectively and thereby introduce more effective techniques combating frequency shortcuts. Meanwhile, the robustness of the metric for frequency shortcut evaluation is yet to be explored in the future.

# 7    Acknowledgement

The work was supported by the National Natural Science Foundation of China under grants no. 62276170, 82261138629, 62306061, the Science and Technology Project of Guangdong Province under grants no. 2023A1515011549, 2023A1515010688, the Science and Technology Innovation Commission of Shenzhen under grant no. JCYJ20220531101412030, Guangdong Provincial Key Laboratory under grant no. 2023B1212060076.

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

# A Appendix

## A.1 More Experiment Results

In this section, we provide more experiment results in cross-domain classification tasks and gradients w.r.t. amplitude to analyze model's learning behavior on different frequency components.

### A.1.1 Detailed Classification Performances

We provide detailed single domain generalization performances in Digits, PACS and CIFAR-10-C in Tabs. 5, 6 and 7, respectively.

Table 5: Experimental results on Digits dataset. Models are trained on the MNIST and results are reported on the listed domains. The best results are labeled in bold.

| Method | USPS | MNIST-M | SVHN | SYNTH | Avg. |
|--------|------|---------|------|-------|------|
| ERM | 76.90 | 52.74 | 27.85 | 39.65 | 49.29 |
| CCSA | 83.72 | 49.29 | 25.89 | 37.31 | 49.05 |
| JiGen | 77.16 | 57.80 | 33.81 | 43.79 | 53.14 |
| AugMix | 80.24 | 75.86 | 63.85 | 69.84 | 72.45 |
| GUD | 77.26 | 60.41 | 35.51 | 45.32 | 55.67 |
| M-ADA | 78.53 | 67.94 | 42.55 | 48.95 | 59.49 |
| RandConv | 84.37 | 87.77 | 57.56 | 62.85 | 72.88 |
| L2D | 83.95 | 87.32 | 62.85 | 63.72 | 74.45 |
| SADA | 89.29 | 75.61 | 68.45 | 72.90 | 76.56 |
| SADA+AugMix | **89.34** | 75.74 | 68.34 | 72.10 | 76.38 |
| Ours | 89.32 | **82.74** | **73.58** | **79.92** | **81.39** |

Table 6: Experimental results on PACS dataset, where the listed domain is adopted for training and the reported results are evaluated on the remaining three domains. PT, AP, CT, SC denote the four domains of photo, art painting, cartoon and sketch, respectively.

| Method | PT | AP | CT | SC | Avg. |
|--------|------|------|------|------|------|
| ERM | 33.52 | 57.86 | 67.84 | 25.12 | 46.09 |
| CCSA | 42.77 | 61.89 | 67.46 | 26.43 | 51.08 |
| JiGen | 43.49 | 63.66 | 70.08 | 32.47 | 52.43 |
| AugMix | 48.27 | 72.92 | 73.81 | 54.88 | 62.47 |
| GUD | 45.62 | 69.47 | 73.46 | 41.67 | 57.56 |
| M-ADA | 48.22 | 70.46 | 75.67 | 43.26 | 59.40 |
| RandConv | 50.86 | 75.82 | 75.46 | 48.90 | 62.76 |
| L2D | 51.17 | 76.90 | 77.80 | 53.68 | 64.74 |
| FACT | 42.7 | 69.7 | 75.2 | 48.9 | 59.1 |
| DSU | 39.10 | 63.80 | 73.60 | 38.20 | 53.70 |
| EFDMix | 48.00 | 75.30 | 77.40 | 44.20 | 61.20 |
| SADA | 51.18 | 77.68 | 76.35 | 57.61 | 65.71 |
| SADA+AugMix | 51.22 | **77.82** | 76.94 | 57.76 | 66.18 |
| AdvStyle | 45.50 | 67.80 | 74.50 | 47.20 | 58.70 |
| Ours | **58.41** | 75.71 | **78.59** | **58.92** | **67.91** |

### A.1.2 Gradients w.r.t. Amplitude

We show gradients of models in the first ten epochs of training in Tab. 8. As shown in Tab. 8, models trained with our generated augmented samples concentrate on learning medium and high-frequency components, while baseline training protocol induces models to learn on low-frequency components.

Table 7: Experimental results on CIFAR-10-C dataset, where the models are trained on CIFAR-10 and evaluated on images with the listed corruption categories.

| Method | Weather | Blur | Noise | Digits | Avg. |
|---|---|---|---|---|---|
| ERM | 67.21 | 56.73 | 30.26 | 62.30 | 54.08 |
| CCSA | 67.66 | 57.81 | 28.37 | 61.96 | 54.04 |
| JiGen | 67.20 | 58.06 | 30.37 | 62.05 | 54.43 |
| AugMix | 78.53 | 82.04 | 64.45 | 76.17 | 75.28 |
| GUD | 69.94 | 60.57 | 48.66 | 60.37 | 59.91 |
| M-ADA | 75.54 | 63.67 | 54.21 | 65.10 | 64.65 |
| RandConv | 76.87 | 55.36 | 75.19 | 77.51 | 71.23 |
| L2D | 75.98 | 70.21 | 73.29 | 72.02 | 72.88 |
| SADA | 79.44 | 80.68 | 70.77 | 78.42 | 77.33 |
| SADA+AugMix | 79.14 | **82.38** | 71.42 | **78.38** | 77.83 |
| Ours | **80.42** | 81.73 | **73.68** | 77.50 | **78.33** |

Table 8: Mean of the absolute value of backward gradients on different frequency components of amplitude spectrum maps in the first 10 epochs. LF, MF and HF denote low, medium and high-frequency components. A larger value represents that more gradients are used to learn on the specific frequency components. A pretrained ResNet-18 is adopted.

| Domain | Data | LF | MF | HF | $\frac{MF}{LF}$ | $\frac{HF}{LF}$ |
|---|---|---|---|---|---|---|
| [c]Art painting | Benign | 7.31e-06 | 1.47e-06 | 7.01e-07 | 0.20 | 0.01 |
| | Ours | 1.76e-06 | 4.79e-06 | 2.34e-06 | **2.72** | **1.33** |
| [c]Cartoon | Benign | 5.99e-06 | 8.65e-07 | 4.02e-07 | 0.14 | 0.07 |
| | Ours | 1.13e-06 | 2.40e-06 | 1.20e-06 | **2.12** | **1.06** |

## A.2   More Visualizations

In this section, we provide more visualizations on feature manifold, t-SNE visualizations, generated augmented samples and dominant frequency maps (DFM) filtered samples.

## A.3   Feature Manifold Visualization

To intuitively show the augmentation space of different methods, we provide feature manifold visualization of images from the source domain with different augmentation variants. As shown in Fig. 4, both the proposed AAD and AAUA explore a border augmentation space compared with the conventional in-batch mixing method [41, 38].

### A.3.1   t-SNE Visualizations

To study the distributions of features for the source and target domains, we provide more t-SNE visualizations of model trained with proposed methods in Fig. 5. One can see that large overlapping areas exist between the source and the unseen target domains, indicating the superiority of the proposed method in exploring wider augmentation space.

### A.3.2   Generated Augmented Samples

To give an intuitive perception of how our approach affects the images, we provide visualizations of benign images drawn from the PACS dataset and their corresponding augmented samples generated by AAUA and AAD in Fig. 6.

### A.3.3   DFM-filtered Samples

In this section, we provide benign samples drawn from ImageNet-10 and their corresponding ones with DFM-filtered out in Fig. 7.

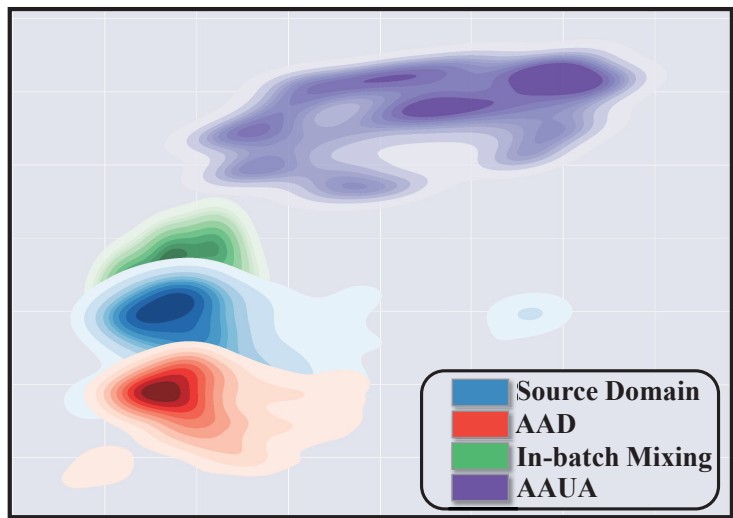

Figure 4: Feature manifold of images from source domain, AAD, in-batch mixing [41, 38] and AAUA. t-SNE [32] is used in this visualization.

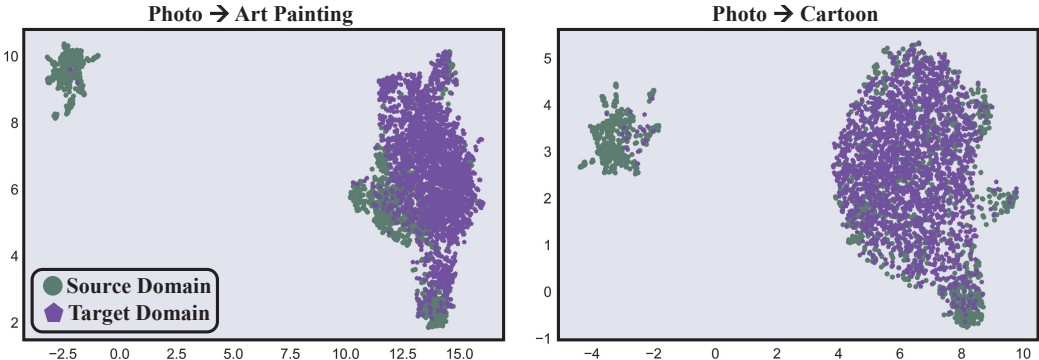

Figure 5: t-SNE visualizations of features from the source domain (Photo) and the unseen target domains (Art Painting and Cartoon) extracted by model trained with our method.

## A.4 Implementation Details

In the task of cross-domain image classification, for Digits, we train a convolution network with the same architecture in [49] with a SGD optimizer for 60 epochs. The initial learning rate is 0.001, which decays by the ratio of 0.1 for every 20 epochs. All the grey images are converted into RGB images. For PACS [19], we train an ImageNet pretrained ResNet-18[9] on the source domain with an Adam optimizer for 60 epochs. The initial learning rate is 0.0001, which follows a cosine annealing decay strategy. For CIFAR-10-C, we train a Wide Residual Network[42] with a width factor of 4 and 16 layers following previous works[10]. The network is optimized with a SGD optimizer for 120 epochs, with an initial learning rate of 0.1 which linearly decays by 0.1 every 40 epochs. Models are trained on CIFAR-10[18] dataset and evaluated on CIFAR-10-C dataset with corruption severity level of 5. All the experiments are run on a Nvidia A100 80GB.

In the task of cross-domain instance retrieval, we adopt the OSNet[48] pretrained on ImageNet, following experiment settings in [45, 44]. The network is optimized with an Adam optimizer for 60 epochs, with an initial learning rate of 0.0003 which linearly decays by 0.1 every 20 epochs.

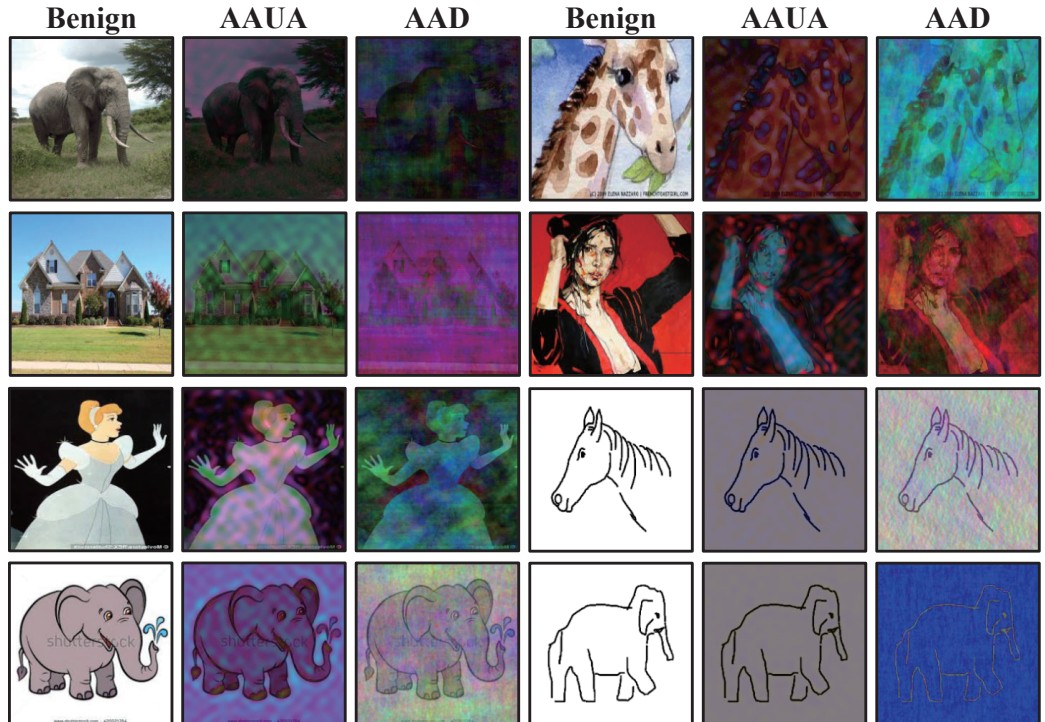

Figure 6: Images from PACS and their corresponding augmented samples generated by AAUA and AAD.

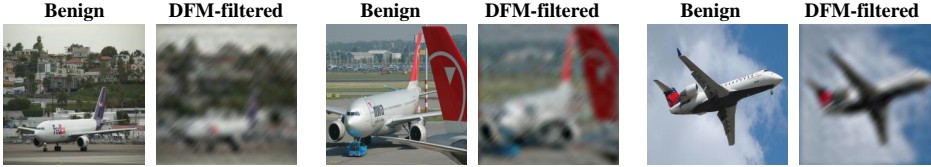

Figure 7: Visualizations of DFM-filtered samples.

## A.5 Frequency Sensitivity Maps

In this section, we discuss how to obtain model's frequency sensitivity maps. We utilize gradients as a surrogate of model's sensitivity to different frequency bands, following:

$$G = \sum_{i=1}^{N} \nabla_{A(x^{(i)})} \mathcal{L}_{task}(x^{(i)}, y^{(i)}; f) \tag{13}$$

where $A(\cdot)$, $N$, $x^{(i)}$ and $y^{(i)}$ denote the obtained amplitude spectrum map, number of images in the dataset, the $i$-th image from the datset and its label, respectively. $G \in \mathbb{R}^{H \times W \times 3}$. We then calculate the mean of $G$ across the channels, following :

$$\bar{G} = \frac{1}{3} \sum_{c=1}^{3} G(:, :, c) \tag{14}$$

We further scale the absolute value of $\bar{G}$ element-wise to the range between 0 and 1 to obtain the frequency sensitivity maps:

$$M_{FS} = s(|\bar{G}|) \tag{15}$$

where $s$ is the min max mapping, which is defined in Eq. (10) of the main body, and $M_{FS}$ denotes the frequency sensitivity map.

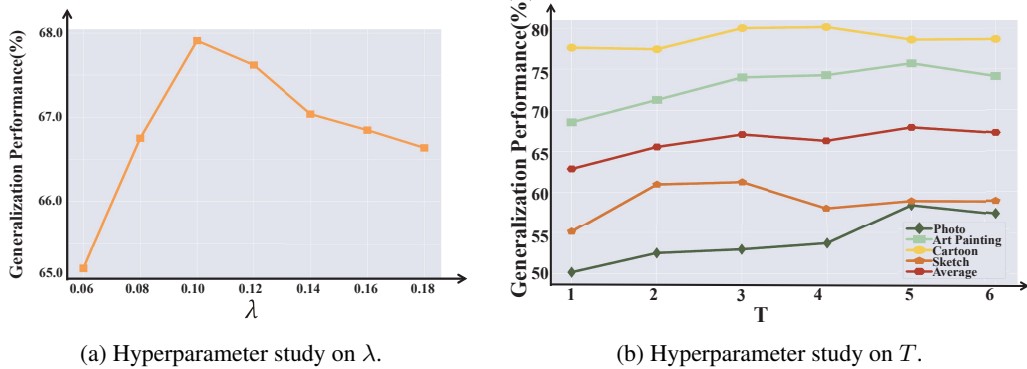

(a) Hyperparameter study on $\lambda$.  (b) Hyperparameter study on $T$.

Figure 8: (a) Generalization performances against different hyperparameter settings of $\lambda$ in AAUA. (b) Hyperparameter analysis of the number of iterative optimization steps, i.e. $T$ in AAUA.

## A.6 Frequency Contribution

In this section, we discuss how to obtain the quantity of frequency contribution (FC). Specifically, FC is formulated as the normalization of $\bar{G}$ in Eq. (14) as:

$$\bar{G}_n = \frac{|\bar{G}|}{\sum |\bar{G}|} \tag{16}$$

where $\bar{G}_n$ denotes the frequency contribution of the whole amplitude spectrum map. To further obtain the low, medium and high frequency contributions, masks for extracting corresponding frequency components are formulated as follows

$$M_{low}(i,j) = \begin{cases} 1 & \text{if} \quad (i^2 + j^2)^{\frac{1}{2}} \leq D_{low} \\ 0 & \text{otherwise} \end{cases} \tag{17}$$

$$M_{mid}(i,j) = \begin{cases} 1 & \text{if} \quad D_{low} < (i^2 + j^2)^{\frac{1}{2}} \leq D_{mid} \\ 0 & \text{otherwise} \end{cases} \tag{18}$$

$$M_{high}(i,j) = 1 - M_{mid}(i,j) - M_{low}(i,j) \tag{19}$$

where $M_{low}$, $M_{mid}$ and $M_{high}$ denote the masks specific to the low, medium and high frequency components, respectively. $D_{low}$ and $D_{mid}$ denote the distances for distinguishing different frequency components. Since the area of low frequency components is relatively small and to obtain equal size of medium and high frequency areas, $D_{low}$ and $D_{mid}$ are set as 10 and 90 for PACS. Then the frequency contribution is obtained by:

$$\begin{cases} FC_{low} = \dfrac{M_{low} \odot \bar{G}_n}{\bar{G}_n} \\ FC_{mid} = \dfrac{M_{mid} \odot \bar{G}_n}{\bar{G}_n} \\ FC_{high} = 1 - FC_{mid} - FC_{low} \end{cases} \tag{20}$$

where $FC_{low}$, $FC_{mid}$ and $FC_{high}$ denote frequency contributions of low, medium and high frequency components, respectively.

## A.7 Hyperparameter Analysis

### A.7.1 Low Frequency Mask Control Factor $\lambda$ in AAUA

In this section, we conduct hyperparameter analysis on $\lambda$, which controls the size of the low frequency mask in AAUA. Experiment results are shown in Fig. 8(a). As $\lambda$ increases, generalization performances first increase. When the perturbated area continues to expand, the injected adversarial noises could harm the stability of the generalized medium frequency components, leading to performance degradation.

### A.7.2 Iterative Times $T$

In this section, we analyse the influence of the number of iterative optimization steps, i.e. $T$ in AAUA on model's generalization performances. As shown in Fig. 8(b), as $T$ increases, the generalization performance increases and then stays stable. Meanwhile, one can see that our AAUA is not sensitive to this hyperparameter setting.

