# OpenReview forum: "Towards Combating Frequency Simplicity-biased Learning for Domain Generalization"
_NeurIPS.cc/2024/Conference — NeurIPS 2024 poster_

### Official Review · Reviewer_szDc · 2024-07-01

**Soundness:** 2
**Presentation:** 2
**Contribution:** 3
**Rating:** 5
**Confidence:** 3

**Summary:**

This paper proposed an augmentation technique in frequency domain, motivated by the phenomenon of frequency bias/frequency shortcut learning. The technique AAUA adversarially perturbs frequency components of images  which the models over-rely for classification. As AAUA might encourage shortcut learning in high-frequency, the authors further use AAD to drop out randomly frequency components that models highly depend on for classification. AAUA and AAD together can avoid frequency shortcut learning. It is shown that AAUA and AAD together can improve domain generalization capability of models.

**Strengths:**

1. Well-written introduction. The introduction explains well the domain generalization problem and how it related to frequency bias / frequency shortcut learning.
2. Interesting and insightful ablation study on the impact of AAUA and AAD on frequency shortcut learning. The authors observed that AAUA or AAD solely might encourage more shortcut learning, supporting the necessity to apply both AAUA and AAD together.

**Weaknesses:**

1. The first and second contribution points can be merged for a compact presentation. Moreover, whether this is the first work to combat frequency shortcuts for domain generalization is uncertain. Domain generalization is a broad topic, and it includes single-source domain generalization problem, e.g. training models on a clean dataset and evaluate them in a corrupted dataset. And the [1] focuses on reducing frequency shortcut learning to improve model robustness towards image corruptions, one of domain generalization problems.

2. As this work proposed a frequency augmentation technique, there are more related work despite FACT and SADA mentioned in the paper, such as VIPAug [2], AFA [3], HybridAugment++ [4], etc.

3. Vague statement: line 327-328: it is uncertain how the comparison is operated given the condition 'without the proposed AAD or AAUA module'. One has to calculate on their own to figure out the comparison is between AAD+L_JS and AAD+AAUA+L_JS, and AAUA+L_JS and AAD+AAUA+L_JS.

4. Sec. 3.2 Theoretical justification is from others' work, motivating the design of the techniques but not having direct mathematical relationship with the proposed technique.  The input-input correlation matrix is introduced but not used.  Strictly speaking, Sec. 3 is more like a literature review explaining motivations instead of analysis carried out by the authors.

5. Definition of the masks separating low and high frequency components: h_min of M_low is H/4, meaning that for frequency components close to the center will be considered as high-frequency, conflict to what is considered as low-frequency. This needs more clarification on why h_min is not 0. Same for w_min.  From the definition of M_low, AAUA seems to shift models' attention to partial mid-high frequencies.

6. Experiments need to be comprehensive:

*As AAUA and AAD augment images in the frequency domain, comparison with more other frequency augmentation techniques should be included, such as HybridAug++ [4], VIPAug [2], AFA [3], DFM-X [1], AugSVF [5].

*The experiments are limited to small datasets, with a few classes. It lacks comprehensive experiments on large datasets, e.g. ImageNet-1k with 1000 classes, to show the feasibility of the proposed technique as well as a fair comparison with other methods which provide results on ImageNet-1k.

*Some results in Table 1 are missing and there lacks explanations.

*In Sec. 3.1 the authors claimed that they focus on classification tasks. However, they abruptly show results on instance retrieval. It is understood that domain generalization is important to instance retrieval. But there needs more clarifications on how the proposed technique can address frequency shortcut learning in retrieval tasks when there are not many work analyzing shortcut learning in image retrieval.

*Sec. 5.3 evaluates frequency shortcuts of trained models. Intuitively, the analyzed models are those trained in Sec. 5.2. However, the evaluation is carried out on ImageNet-10, which is not mentioned previously and not used for training. It is unclear at this stage whether the authors use the models trained in Sec. 5.2 for shortcut evaluation on the dataset ImageNet-10, or train new models on ImageNet-10 for shortcut evaluations. This needs clarifications.

*Sec. 5.4.1 demonstrates that the proposed augmentation technique increases the frequency sensitivity of models compared to baseline. This seems to be a bad sign to model generalization capability according to [43]. However, there is no related discussions and Fig. 3a is not referred anywhere in the main paper.

*Sec. 5.4.2 analyzes the hardness of augmented samples. However, the comparison is between AAUA and FACT, which seems unfair as FACT does not include adversarial setup while AAUA does.

Minor:
Each formula should end with period/comma. Line 271 'relative' should be 'related' if understood correctly. Table 3 not in the same page where mentioned. Table 4 exceeds width limit.

[1] Wang, et al., 'DFM-X: Augmentation by Leveraging Prior Knowledge of Shortcut Learning', ICCVW2023.

[2] Lee, et al., 'Domain Generalization with Vital Phase Augmentation', AAAI2024.

[3] Vaish, et al., 'Fourier-basis Functions to Bridge Augmentation Gap: Rethinking Frequency Augmentation in Image Classification', CVPR2024.

[4] Yucel, et al., 'HybridAugment++: Unified Frequency Spectra Perturbations for Model Robustness', ICCV2023.

[5] Soklaski, et al., 'Fourier-Based Augmentations for Improved Robustness and Uncertainty Calibration', Neurips2021.

**Questions:**

1. How are d_t^mu and the other initialized? Did the experiments consider different random seeds?

2. Speaking of hardness of the augmented samples, how do AAUA and AAD compare with other adversarial techniques?

3. Line 121-123: If correctly understood, it should be: responsive frequency describes the characteristics of mapping functions learned by networks and the mapping function transforms images into probability vectors. How does the smoothness represented by the probability vectors and how does it relate to frequency? The concept of 'smoothness' appears abruptly.

4. What are the meaning of the two target domains in Fig. 2? Are they to show the distributions of augmented images are close to them? How are the density distributions of each domain computed? What do the peak values of each distribution mean?

**Limitations:**

The authors addressed the limitations of their methods, regarding the infeasibility to directly locate frequency shortcuts.

---

> ### Author Rebuttal · Authors · 2024-08-06
>
> ## Merging Contributions
>
> Thank you for your valuable comments. We will merge the first two points as suggested.
>
> ## Claim as The First
>
> **(1) [General Robustness vs. Domain Generalization(DG)]** We acknowledge the correlation between general robustness studies (i.e. DFM-X) and DG, but respectfully highlight that DG involves stronger distribution shifts and semantic variances, compared to simply training on clean data and testing on its corruption subset.
>
> **(2) [Specific Focus on Frequency Shortcuts]** We evaluate frequency shortcut mitigation and improved DG performance, offering a novel contribution compared to DFM-X, which uses learned frequency shortcuts for augmentation but lacks a comprehensive evaluation of frequency shortcut defending.
>
> **(3) [Further Experiments]** As shown in the latter, the failure of the recommended papers (mostly work on general robustness) on single source DG, underscores the distinction between these two challenges. If general robustness were equivalent to DG, these papers would not consistently fail on the DG dataset. This suggests that general robustness and DG are indeed distinct tasks.
>
> **(4) [Further Modification]** We are aware that there might be potential uncertainty, and we would replace the “first work” with “one of the pioneering works”.
>
> ## Comparison with More Papers
>
> **(1) [Additional Experiments]**  We compare with recommended papers on PACS(DG), ImageNet-1k(general robustness), and ImageNet-10(frequency shortcut). Results in Tab. Re9,Re10,Re11 show our method's competitiveness, outperforming others on PACS and ImageNet-10 while remaining competitive on ImageNet-1k.
>
> **(2) [Reason of Dataset Scope]** We didn't compare these methods and on ImageNet because our focus is on DG, with relevant methods (FACT and SADA) and established datasets. Our dataset scope aligns with established single-source DG papers[43,36,6,3].
>
> **(3) [Large-scale DG dataset]** We've added results on a larger-scale DG dataset: DomainNet, showing superior performance (Tab. Re10).
>
> ## Statement on Line 327-328
>
> We are sorry for the confusion caused. We will add a performance degradation term in the ablation table for clarity so that one can easily understand the comparison with detailed numbers, as shown in Tab. Re12.
>
> ## Theory Section
>
> **(1)** The theory section is to bridge the gap between the theoretical proof in [24] and the frequency shortcut paper[35]. While [24] doesn't mention frequency shortcuts and [35] is purely empirical, we connect these works (Line:160-173), motivating our method. We've correctly cited [24] for the theory.
>
> **(2)** We will remove the spare input-input correlation matrix, while it is represented by "the dataset statistical structure" in the paper.(i.e. Line:151 and Line:191).
>
> **(3)** The matrix is calculated on the entire dataset. Yet, we can't backward on the entire dataset in one back propagation (due to GPU memory), thus we don’t directly use it and propose our in-batch technique.
>
> ## Mask in AAUA
>
> **(1)** We clarify that the lowest frequency is shifted to the center after FFT. Thus, frequency in coordinate (2/H, 2/W) is the lowest frequency and higher frequencies are farther from this center point.
>
> **(2)** AAUA alone would shift the model's attention to partial mid-high frequencies(Line:239-240). This is based on the intuition that low frequencies dominate real-world datasets and are more likely to hide frequency shortcuts.
>
> ## Comprehensive Experiments
>
> **[Missing Results]** Results in Tab. 1 are drawn from the corresponding papers, the missing ones indicate they didn’t run experiments on the corresponding datasets.
>
> **[Shortcut Learning in Re-ID]** The re-ID model is trained with the same loss as classification (each person has a corresponding ID), resulting in the same optimization targets. We'll provide a brief explanation of re-ID, but omit detailed analysis due to its overlap with classification.
>
> **[Frequency Shortcut Evaluation]** We follow the protocol in [35] for shortcut evaluation, where the models are trained on ImageNet-10. We will further clarify it.
>
>  **[Sensitivity Map in Fig. 3]**  **(1) [Different Definition]** We kindly emphasize that sensitivity maps in [43] are different from ours. They are accumulation errors on the source domain with Fourier-basis noise, while ours are derived from gradients (as shown in Appendix A.5).  **(2) [Uncertain Claim]** [43] only provides sensitivity maps on the source domain. The claim "lower sensitivity, better generalization" remains uncertain for target domains (conclusion section of [43]). **(3) [Additional Figure]** Additional sensitivity maps (Fig. Re3) following [43], show that our method suppresses source domain sensitivity [43], indicating good generalization. Fig.3(a) would be cited in Sec.5.4.1
>
>  **[Hardness Comparison]** Adversarial is a core component of our method. Thus, the comparison is fair. Fig. Re2 shows that AAUA covers a larger range than adversarial SOTAs (ABA[3] and ALT[6]), demonstrating higher hardness.
>
>  **[Initialization and Random Seeds]** $d_t^{\mu}$ and others are randomly initialized with a normal distribution N(0,1). Experiments are run over three different random seeds.
>
> ## "Smoothness" and Responsive Frequency
>
> High responsive frequency means that small input changes would significantly affect the model output (i.e. adversarial attack). As this concept is closely related to the Lipschitz constant, which describes the model's smoothness, we thus use the term.
>
> ## Manifold Map
>
> **(1)** We assume that you are referring to Fig. 3(b). The target domains demonstrate our effectiveness in crafting augmentation domains, rather than suggesting the augmented images are close to the target ones, as they are unseen in DG. **(2)** Density distributions are computed with kernel density estimation based on the t-SNE results. One could do so with the Python package seaborn.kdeplot. Peak values mean more augmented images lie in that point after dimension reduction.

---

> > ### Comment · Reviewer_szDc · 2024-08-11
> >
> > 1. Unclear claim regarding general robustness and experiment scope: The paper and the rebuttal lack a clear definition of "general robustness" and assert that general robustness and domain generalization are distinct problems. This conclusion is drawn from comparing methods designed for general robustness within domain generalization tasks. However, since general robustness could be viewed as a subset of domain generalization, particularly as a single-source domain generalization problem, it's expected that methods designed for general robustness might underperform compared to those developed for the broader problem. Therefore, experimenting on ImageNet and its corruption and generalization variants is reasonable. The authors also provided results on CIFAR-C, a corruption dataset for 'general robustness', instead of DG problem.
> > 2. Limited evaluation of frequency shortcut mitigation: The evaluation of frequency shortcut mitigation is conducted only on ImageNet-10, which limits the range of the assessment. It’s unclear how well the proposed method addresses frequency shortcuts across other datasets, and a broader evaluation is needed to provide a more comprehensive understanding of the method’s effectiveness.
> > 3. Intuition of AAUA's attention on mid-high frequencies: The paper suggests that AAUA focuses on mid-high frequencies because low frequencies are more likely to contain frequency shortcuts. However, this claim lacks sufficient support, as the referenced paper [35] does not make this claim. More evidence or clarification is needed to strengthen this reasoning.
> > 4. Missing experiments for comprehensive comparison: Some key experiments from related work are not included in the paper, which limits the comparison. Running these experiments or explaining why they were omitted  is necessary for the comprehensiveness of this study. Additionally, the lack of large-scale experiments on ImageNet is a noticeable gap that should be addressed.
> > 5. Use of gradients in sensitivity map: The paper uses gradients for the sensitivity map instead of accumulated errors, as in [43]. The authors suggest that accumulated errors might not indicate better generalization for target domains, but the paper and the rebuttal do not clearly explain how this issue is resolved. More explanation is needed to justify this approach and what the proposed sensitivity map represents.
> > 6. Hardness comparison and method evaluation: The adversary nature is a core component of the proposed adversarial method, but comparing it with non-adversarial methods is not entirely fair. Moreover, while AAUA covers a broader range than ABA, the performance improvement is minimal (as shown in Table Re 1). This raises questions about whether hardness is a sufficient metric for comparison, and additional factors may need to be considered.

---

> > > ### Author Response · Authors · 2024-08-12
> > > **Discussion Post 1**
> > >
> > > ### General Robustness and DG
> > >
> > > **(1)** General robustness studies often engage with model robustness against simple distribution shifts, adversarial attacks and common corruptions [D1,D2,D3]. In general robustness, they typically focus on variations within the same domain (e.g., noise, adversarial attacks)[D1,D2,D3]
> > >
> > > **(2)** DG often considers stronger distribution shifts and semantic variances, where the training and testing data are considered as different domains. As such, DG [22,43,36,6,3,46,39,37,38] and general robustness [D1,D2,D3] are mostly different research communities.
> > >
> > > **(3)** Prior works on single-source DG never conducted experiments on ImageNet, but instead provided results on PACS, Digits and CIFAR-10-C[22,43,36,6,3,46,39,37,38]. We are following the evaluation protocol from these prior works.
> > >
> > > #### References:
> > >
> > > D1: Subbaswamy, Adarsh, Roy Adams, and Suchi Saria. "Evaluating model robustness and stability to dataset shift." International conference on artificial intelligence and statistics. PMLR, 2021.
> > >
> > > D2: Yin, Dong, et al. "A fourier perspective on model robustness in computer vision." Advances in Neural Information Processing Systems 32 (2019).
> > >
> > > D3: Liu, Chang, et al. "A comprehensive study on robustness of image classification models: Benchmarking and rethinking." International Journal of Computer Vision (2024): 1-23.
> > >
> > > ### Evaluation on Frequency Shortcut
> > >
> > > **(1)** Regarding the question on the effectiveness of evaluation on frequency shortcuts, we kindly emphasize that we strictly follow the frequency shortcut evaluation protocol from [35], which it evaluates on ImageNet-10. To the best of our knowledge, there's no existing literature conducting frequency shortcut evaluation on any other datasets except ImageNet-10.
> > >
> > > **(2)** Given the time-consuming process of masking each frequency band separately and then evaluating accuracy per class, it is not feasible to conduct frequency shortcut evaluation on the ImageNet-1k in the relatively short rebuttal period.
> > >
> > > ### Intuition of AAUA
> > >
> > > **(1)** In the theory section, one would notice that frequency shortcuts are more likely hidden in the dominant frequencies. Meanwhile, in real-world datasets, the amplitude of a natural image is often dominated by low-frequency components.
> > > In this case, we thus apply AAUA to the low frequencies.
> > >
> > > **(2)** [35] is a fully empirical study and they indeed did not provide such a claim. However, the reason we apply AAUA to low-frequencies is mainly analyzed from the theory section and the intuition of low-frequencies being dominant in natural images. The absence of such a statement in [35] does not imply that our intuition is incorrect.
> > >
> > > ### Some Missing Results in Tab.1
> > >
> > > **(1)** To ensure a fair comparison with other methods, we directly report the results in Tab.1 from the corresponding papers. As stated in the rebuttal, missing results in the experiments indicate the corresponding papers didn't provide results on the corresponding datasets. It would not be possible to run all these methods ourselves during the limited time of the rebuttal period. We will complete them in the next version of the paper.
> > >
> > > **(2)** We've already conducted extensive additional comparisons in the rebuttal, as shown in Tab. Re1,Re3,Re8,Re9,Re12.
> > >
> > > ### Large-scale Experiments
> > >
> > > **(1)** We have already provided large-scale experiments on ImageNet and DomainNet in Tab. Re10 and Tab. Re12 in the rebuttal PDF, respectively.
> > >
> > > **(2)** To the best of our knowledge, existing DG papers [22,43,36,6,3,46,39,37,38] haven't conducted experiments on ImageNet. Conversely, they have the same dataset scope as ours (PACS, Digits, CIFAR-10-C). We are strictly following this protocol from previous DG works.
> > >
> > > ### Sensitivity Map
> > >
> > > **(1)** Our sensitivity map with gradients is followed from [2], we are sorry for using the wrong citation in Sec. 5.4.1 and thus causing misunderstanding. Our sensitivity map shows the model's preferences on different frequency components when giving predictions [2].
> > >
> > > **(2)** The claim that "accumulated errors might not indicate better generalization for target domains" is not mentioned in the paper, because the sensitivity map in the main paper is completely different from that of [43]. We claim this in the rebuttal because we show that our gradient sensitivity map in the paper and models on different domains exhibit different sensitivity and we intend to avoid any confusion regarding our sensitivity maps. As our primary focus is DG, we thus do not investigate differences between our sensitivity map and that from [43].
> > >
> > > #### References: (In main paper)
> > >
> > > [2] Alvin Chan, Y. Ong, and Clement Tan. How does frequency bias affect the robustness of neural image classifiers against common corruption and adversarial perturbations? In International Joint Conference on Artificial Intelligence, 2022

---

> > > ### Author Response · Authors · 2024-08-12
> > > **Discussion Post 2**
> > >
> > > ### Hardness Comparison
> > >
> > > **(1)** Regarding the new question of whether hardness comparison is a sufficient metric, the hardness comparison map only serves as a qualitative visualization result to compare the strength of the augmentations. We kindly request the reviewer if he/she could provide any other metric to help further improve this analysis.
> > >
> > > **(2)** Similar hardness comparisons are also provided in the supplementary material of ABA[3] and the main papers of ALT[6] and AdvST[46].
> > >
> > > **(3)** On the other hand, we've never stated that learning hardness is highly correlated with generalization performances. Our primary focus is domain generalization performance, and so whether hardness is a sufficient metric for domain generalization is beyond our scope.
> > >
> > > ### Performance Improvement in Tab. Re1
> > >
> > > Results in Tab. Re1 shows that our method achieves the shortest additional training time and the best DG performances among the adversarial augmentation SOTAs[3,6,46]. We outperform the SOTA ABA with a clear margin of 1.89\% in terms of performance. Note that we achieve the best performance with the shortest additional training time (about $\frac{1}{5}$ of ABA).
> > >
> > >
> > >
> > > We hope that, our responses have adequately resolved reviewer's concerns. We would kindly request the reviewer to re-consider improving the rating.

---

> ### Comment · Reviewer_szDc · 2024-08-12
>
> 1.General Robustness and Domain Generalization (DG): The connection between corruption robustness and DG is discussed in the survey paper (https://ieeexplore.ieee.org/stamp/stamp.jsp?arnumber=9847099), which suggests that Out-of-Distribution (OOD) robustness can be seen as a type of single-source DG problem. You mentioned that "DG considers stronger distribution shifts and semantic variances." However, strong distribution shifts are also relevant in cases of corruption and adversarial attacks. Moreover, your statement that "DG [22,43,36,6,3,46,39,37,38] and general robustness [D1,D2,D3] are mostly different research communities" lacks clear criteria for measuring the magnitude of distribution shifts or semantic variances. There doesn't seem to be solid evidence supporting the idea that DG involves stronger semantic variances. In many DG studies, the term "domain" often just refers to different environments for data collection, without necessarily indicating strong semantic variance.
>
>
> 2. Evaluation on Frequency Shortcut Mitigation: Given the title "Combating Frequency Simplicity-biased Learning," it is crucial to demonstrate that the proposed method effectively mitigates frequency shortcuts. Therefore, a more comprehensive evaluation is necessary to validate the benefits of your approach in this context.
>
>
> 3. Intuition Behind AAUA: The explanation provided for AAUA’s focus is somewhat unclear. The response states that "frequency shortcuts are more likely hidden in the dominant frequencies," and adds that "in real-world datasets, the amplitude of a natural image is often dominated by low-frequency components." However, it’s not entirely clear what is meant by "dominant frequencies"—whether it refers to those dominating amplitude or classification. Furthermore, according to paper [35], the dominant frequencies in classes that use frequency shortcuts (as seen in their DFMs) often include many mid-high frequencies, rather than just low frequencies.

---

> > ### Author Response · Authors · 2024-08-12
> > **Discussion Post 3**
> >
> > Thank you for taking time to respond to our comments.
> >
> > 1. We kindly note that the discussion has now been drifting away from the scope of the paper. We previously talked about this in rebuttal to try to address the reviewer's concern on the "first claim" and "ImageNet results". As mentioned in the rebuttal, we would replace the "first claim" with "one of the pioneering works" and we've provided ImageNet-1k results following the reviewer's suggestions.
> >
> > 2. Following your suggestion, we further provide frequency shortcut evaluation on other datasets (PACS and CIFAR-10) below. The experiment results below further demonstrates the effectiveness of the proposed method in frequency shortcut mitigation.
> >
> > | **Method** | **PACS (Avg.TPR$\downarrow$ / Avg.FPR $\downarrow$)** | **CIFAR-10(Avg.TPR$\downarrow$ / Avg.FPR $\downarrow$)** |
> > | ---------- | ----------------------------------------------------- | -------------------------------------------------------- |
> > | ResNet-18  | 0.444 / 0.197                                         | 0.423 / 0.130                                            |
> > | AFA        | 0.426 / 0.079                                         | 0.441 / 0.106                                            |
> > | VIP        | 0.455 / 0.133                                         | 0.416 / 0.127                                            |
> > | HAP        | 0.402 / 0.153                                         | 0.397 / 0.092                                            |
> > | Ours       | **0.311 / 0.047**                                     | **0.258 / 0.073**                                        |
> >
> > 3. Dominant frequencies are the spatial frequencies that contribute most to the singular vectors of the dataset's statistical structure. Low frequencies are generally the dominant ones for natural images because they contain most of the information images carry. Meanwhile, low frequencies are often considered to carry domain-specific information [38]. We thus put AAUA's focus on low-frequencies.

---

> > > ### Comment · Reviewer_szDc · 2024-08-13
> > >
> > > Thank you for the rebuttal. I want to clarify that the discussion regarding DG and 'general robustness' is closely related to experiment design, especially in the choice of datasets. This is why I suggested conducting experiments on ImageNet-1k and its OOD variants (e.g. ImageNet-C-bar, ImageNet-P, ImageNet-A, ImageNet-Sketch, etc.), though I did not list all of them in my initial comments.
> > >
> > > However, I have concerns about the approach of using average TPR and FPR to evaluate frequency shortcuts. The paper [35] that investigates frequency shortcuts conducts its analysis on a class-wise manner rather than relying on average values. Thus, I examined the average TPR and FPR in [35] and compared them with the OOD results in this paper, I noticed some inconsistencies. For example, the model 'ResNet18+SIN,' which has the best average OOD performance, does not have the lowest average TPR on ImageNet-10. Instead, 'ResNet18+AutoAug' shows the lowest average TPR, yet it only achieves average OOD performance compared to other models. This discrepancy suggests that the chosen metric for evaluating frequency shortcuts may not be robust (could be the reason that [35] did not propose it in their paper), and its validity requires further substantiation.
> > >
> > > Based on these considerations, I believe the manuscript is not ready for publication, as it still needs improvements in writing, readability (as noted by Reviewer jMvT) as well as addressing technical concerns. Thus, I will maintain my current rating.

---

> > > > ### Author Response · Authors · 2024-08-13
> > > > **Discussion Post 4**
> > > >
> > > > Thank you for your response.
> > > >
> > > > 1. We follow the protocol in [35] to provide class-wise TPR/FPR results below. It can be seen that the proposed method outperforms the baseline across 8/10, 7/7 and 8/10 classes on ImageNet-10, PACS and CIFAR-10, respectively.
> > > >
> > > > 2. As mentioned in the limitations and future work, existing works (including ours) couldn't directly locate frequency shortcuts, thus it would be hard to measure this with a robust metric. We kindly ask if the reviewer could provide any more specific metrics or suggestions. Meanwhile, whether the TPR/FPR metric we followed from prior work [35] is robust, is beyond the scope of this paper.
> > > >
> > > > 3. In this paper, we focus on preventing frequency shortcuts to improve DG performance. Neither this paper nor [35] stated that the least frequency shortcuts always result into the best DG performances.
> > > >
> > > > ImageNet-10
> > > >
> > > > | Method           | Class 1     | Class 2 | Class 3  | Class 4     | Class 5     | Class 6     | Class 7     | Class 8     | Class 9     | Class 10    |
> > > > | ---------------- | ----------- | ------- | -------- | ----------- | ----------- | ----------- | ----------- | ----------- | ----------- | ----------- |
> > > > | ResNet-18        | 0.08/0.0044 | 0/0     | 0.4/0.02 | 0.8/0.0356  | 0.02/0.0311 | 0.02/0.0044 | 0.14/0.0044 | 0.8/0.1178  | 0.54/0.1889 | 0.06/0.0022 |
> > > > | ResNet-18 + Ours | 0/0         | 0/0     | 0.15/0   | 0.34/0.0465 | 0.08/0.0257 | 0/0.0142    | 0.12/0.0015 | 0.41/0.0612 | 0.24/0.1509 | 0.78/0      |
> > > >
> > > > PACS
> > > >
> > > > | Method           | Class 1    | Class 2     | Class 3     | Class 4     | Class 5     | Class 6     | Class 7     |
> > > > | ---------------- | ---------- | ----------- | ----------- | ----------- | ----------- | ----------- | ----------- |
> > > > | ResNet-18        | 0.325/0.18 | 0.448/0.168 | 0.467/0.157 | 0.522/0.154 | 0.359/0.251 | 0.568/0.264 | 0.417/0.205 |
> > > > | ResNet-18 + Ours | 0.218/0    | 0.223/0.02  | 0.439/0.113 | 0.406/0.146 | 0.269/0.005 | 0.417/0.046 | 0.205/0     |
> > > >
> > > > CIFAR-10
> > > >
> > > > | Method           | Class 1    | Class 2      | Class 3     | Class 4     | Class 5     | Class 6     | Class 7     | Class 8     | Class 9     | Class 10    |
> > > > | ---------------- | ---------- | ------------ | ----------- | ----------- | ----------- | ----------- | ----------- | ----------- | ----------- | ----------- |
> > > > | ResNet-18        | 0.39/0.117 | 0.085/0.0011 | 0.483/0.184 | 0.749/0.303 | 0.856/0.457 | 0.461/0.140 | 0.268/0.023 | 0.254/0.015 | 0.556/0.054 | 0.132/0.003 |
> > > > | ResNet-18 + Ours | 0.29/0     | 0/0          | 0.254/0     | 0.538/0.364 | 0.502/0.127 | 0.155/0.002 | 0.216/0.015 | 0.41/0.132  | 0.209/0.092 | 0.04/0      |

---

> > > > > ### Comment · Reviewer_szDc · 2024-08-14
> > > > >
> > > > > Thank you again for the rebuttal.
> > > > > 1. Concern regarding evaluation
> > > > > As the title suggests, "Combating Frequency Simplicity-biased Learning for Domain Generalization," the authors implicitly hypothesize that reducing frequency shortcut learning will improve domain generalization. Therefore, I think it crucial to evaluate whether the proposed technique effectively reduces frequency shortcut learning. Thus, having an appropriate metric to evaluate this is important. If this is not the authors' intention, they should consider rephrasing their statements in the paper and possibly changing the title to avoid such confusion.
> > > > >
> > > > > 2. Concern regarding motivation
> > > > > The assumption that "the least frequency shortcuts always result in the best DG performances" is inferred from the authors' methodology, where adversarial learning is used to prevent frequency shortcuts, with the aim of achieving the best DG performance. This naturally leaves the impression that the  well-generalized models are expected to exhibit the least frequency shortcut learning. This assumption is closely related to the motivation behind the method's design. If this is not the authors' intention, it needs to be clarified in the paper, and the motivation behind the design should be better explained.
> > > > >
> > > > > If the authors can provide reasonable ideas for addressing the concerns and better formulate their motivation, I would be happy to reconsider my rating.

---

> > > > > > ### Author Response · Authors · 2024-08-14
> > > > > > **Discussion Post 5**
> > > > > >
> > > > > > We deeply appreciate your engagement in the discussion.
> > > > > >
> > > > > > 1. To help address the concerns, we would be happy to incorporate both the average metric and class-wise metric for frequency shortcut evaluation, across the ImageNet-1k and its OOD variants, in the final version of the paper. Further, we would also clarify this in both experiment settings and the discussion of limitations in the final version of the paper.
> > > > > >
> > > > > > 2. Regarding the motivation for adversarial technique, we would like to clarify that the connection between adversarial learning and DG performance is step-by-step, instead of simply aiming for the best DG performance.
> > > > > > As mentioned in Line: 174-179, the learning behavior is deemed to pick up dominant frequencies (Line:150) given a static dataset with no augmentations applied, we thus turn to the side of data-centered method to leverage this learning behavior.
> > > > > > To do so, as mentioned in Line: 180-194, we would have to create data augmentations that are aggressive enough to affect the statistical structure of the dataset.
> > > > > > Given the dynamic learning behavior of a model, one way to create hard samples for the model in each iteration is adversarial learning, we thus propose to apply the adversarial technique.
> > > > > > Meanwhile, the adversarial gradient w.r.t amplitude utilized in AAD could be interpreted as the model's frequency preference when giving predictions [2], thus the adversarial technique is also applied to capture the over-reliance frequencies in each training iteration.
> > > > > >
> > > > > > Once again, we deeply appreciate your engagement in the discussions for multiple rounds and truly hope that our discussion could convince the reviewer for a better rating.

---

### Official Review · Reviewer_57HS · 2024-07-03

**Soundness:** 2
**Presentation:** 2
**Contribution:** 3
**Rating:** 5
**Confidence:** 3

**Summary:**

The paper addresses the challenge of domain generalization by focusing on the issue of frequency simplicity-biased learning in neural networks. This phenomenon leads to an over-reliance on specific frequency sets, known as frequency shortcuts, instead of semantic information, thereby impairing generalization performance. The innovative data augmentation techniques adaptively manipulate the learning difficulty of different frequency components, thereby enhancing domain generalization performance by combating frequency shortcut learning.

**Strengths:**

The paper presents a novel approach to tackling the issue of frequency simplicity-biased learning in domain generalization. The introduction of adversarial frequency augmentation modules, specifically Adversarial Amplitude Uncertainty Augmentation (AAUA) and Adversarial Amplitude Dropout (AAD), is innovative. These methods provide a new perspective by manipulating the dataset's frequency characteristics to prevent over-reliance on specific frequency components, which is a significant departure from traditional data augmentation techniques. This creative combination of adversarial learning with frequency domain manipulation represents a fresh and original contribution to the field.The theoretical foundation provided in the paper is robust, offering a clear justification for the proposed methods. The experimental validation is thorough, with the proposed techniques being rigorously evaluated across various benchmarks, including image classification and instance retrieval tasks. The results demonstrate a substantial improvement over state-of-the-art methods, highlighting the effectiveness of the proposed approach.

**Weaknesses:**

The effectiveness of AAUA and AAD might be sensitive to the choice of hyperparameters, such as the intensity of adversarial perturbations and the threshold for frequency dropout. The paper does not provide an in-depth analysis of the sensitivity of the results to these hyperparameters. The proposed adversarial frequency augmentation methods, AAUA and AAD, involve iterative optimization steps and adversarial gradient computations, which could be computationally expensive.

**Questions:**

Why did you choose the specific baselines for comparison, and how do you think your methods would perform against other recent or diverse approaches?
What is the computational overhead associated with implementing AAUA and AAD?

**Limitations:**

The authors acknowledge the challenge of directly locating frequency shortcuts. Providing preliminary insights or future directions on how to address this limitation would be a constructive addition. This could involve discussing potential methods for identifying and mitigating these shortcuts more directly.

---

> ### Author Rebuttal · Authors · 2024-08-06
>
> ## More Hyper-Parameter Analysis.
>
> **(1) [Additional Results]** Thank you for your valuable comments. Following your suggestion, we further provide hyper-parameter studies of the intensity of AAUA perturbations and the probability threshold $p$ for AAD in Tab. Re6 and Tab. Re7, respectively.
> As we don't have a clamping operation to directly restrict the noise intensity like traditional adversarial attacks [Re1] do, we thus provide an ablation on the update step size of the AAUA, as this would directly control the intensity of AAUA given static iterative steps of 5.
>
> **(2) [Additional Analysis]** As shown in Tab. Re6 and Tab. Re7, the hyper-parameter setting in the main paper generally achieves the best performance. In Tab. Re6, it can be seen that both overly strong and weak perturbation intensity leads to sub-optimal performances. Because an overly strong intensity would produce augmentation domains with excessive domain-shift to learn while an overly weak intensity fails to create enough domain shift. In Tab. Re7, it be seen that a relatively higher threshold leads to better performances, because it could help filter out the frequencies with higher dependency.
>
> ## Computational Overhead Analysis.
>
> **(1) [Additional Experiments]** We agree that the proposed technique, involving adversarial gradient computations, would bring additional computation overhead. We conducted an efficiency study to evaluate running time per epoch and performance improvement over the baseline method (ERM). This study includes our proposed method and several related adversarial augmentation techniques [3,6,46] on the PACS dataset. The experimental results are presented in Tab. Re1. All timings were measured on a single NVIDIA A100-80G GPU with a batch size of 32. The "Photo" domain of PACS was utilized as the training domain.
>
> **(2) [Experiment Analysis]** As illustrated in Tab. Re1, our method achieves the highest performance compared to the state-of-the-art adversarial augmentation techniques [3,6,46], with only a minimal increase in training time over ERM. This demonstrates the superior efficiency of our approach, which we attribute to its parameter-free nature (i.e. ABA has a Bayesian network to help generate augmentation) and the reduced number of iterative optimization steps (Ours:5 steps, AdvST: 50 steps, ALT:10 steps, ABA:14 steps).
>
> ## Reason for Choosing the Specific Baselines and Comparison with Other Methods.
>
> **(1)** Most of the chosen baselines are adversarial augmentation methods and Fourier augmentation methods, which are highly related to ours.
>
> **(2)** Following your suggestion, we further include more diverse methods for comparison in Tab. Re8, showing comparative performances of the proposed method. Due to space limits, we show below the citations to the corresponding works in Tab. Re8.
>
>
>
> ## References:
>
> Re1: Madry, Makelov, et al. "Towards Deep Learning Models Resistant to Adversarial Attacks." ICLR (Poster) 2018
>
> MetaCNN: Chen, Jin, et al. "Meta-causal learning for single domain generalization." *Proceedings of the IEEE/CVF Conference on Computer Vision and Pattern Recognition*. 2023.
>
> Pro-RandConv: Choi, Seokeon, et al. "Progressive random convolutions for single domain generalization." *Proceedings of the IEEE/CVF Conference on Computer Vision and Pattern Recognition*. 2023.
>
> R-XCNorm: Chuah, WeiQin, et al. "Single Domain Generalization via Normalised Cross-correlation Based Convolutions." *Proceedings of the IEEE/CVF Winter Conference on Applications of Computer Vision*. 2024.
>
> PDDOCL: Li, Deng, et al. "Prompt-Driven Dynamic Object-Centric Learning for Single Domain Generalization." *Proceedings of the IEEE/CVF Conference on Computer Vision and Pattern Recognition*. 2024.

---

> ### Comment · Reviewer_57HS · 2024-08-13
>
> I appreciate the author's response; According to the disscusions among all the reviewers' and the overall concerns, I keep the rating.

---

### Official Review · Reviewer_jMvT · 2024-07-15

**Soundness:** 2
**Presentation:** 1
**Contribution:** 2
**Rating:** 5
**Confidence:** 4

**Summary:**

The paper proposes a method for single source domain generalization. The method is based on the insight of countering frequency shortcuts learnt during training. The paper proposes to augment images in the Fourier domain to achieve this. Such augmentation is done adversarially, by changing the mean and variance of low frequency components to maximize the loss on such samples, and by adversarially dropping some frequencies based on their gradients. The method is benchmarked on a variety of classification and retrieval tasks, and superior performance is demonstrated.

**Strengths:**

* Extensive ablation studies are performed on each component of the method.
* Adequate background on the method is provided.

**Weaknesses:**

* The writing seems a bit belabored and hard to follow. The paper's presentation can be helped by a thorough rewrite, focussing on the grammar and readability.
* Some of the implementation and experimental details are not clearly mentioned. See list of questions below.
* Some of the design choices seem arbitrary, and I was not able to follow the motivation for them. See list of questions below.

**Questions:**

* The choice of using AAUA only for low frequency components and AAD mainly for higher frequency seems a bit arbitrary. One could apply AAUA on both components, or only on high frequency components as well. Do the authors have an intuition/justification on whether this would work better?
* What is the JS loss over? Is it consistency between predictions on augmented and clean samples?
* The experimental setup is not clear enough - what are the training domains in Table 1? Was the experiment run multiple times? How were the hyper-parameters for each method chosen (i.e. was there an external validation set?) What is the in-domain accuracy (i.e. does AAUA hurt in-domain performance a lot)?
* Fig 3 is hard to follow. What is the conclusion from Fig 3(a)?
* In Fig.5, what do baseline tSNE plots look like?

**Limitations:**

The authors have adequately addressed their limitations.

---

> ### Author Rebuttal · Authors · 2024-08-06
>
> ## Writing.
>
> Thank you for your detailed comments. We will thoroughly revise the manuscript.
>
> ## Implementation details and clarity of experiment results.
>
> **[JS loss]** Yes, the JS loss is applied between predictions on augmented and clean samples
>
> **[Training domains]**
>
> **(1)** Kindly note that the training domains are illustrated in the captions of Tables. 5,6,7 in the appendix of the submission (Line:494,506). We follow this established protocol from previous works[43,36,6,3,46].
>
> **(2)** For Digits, MNIST is used as the training domain. For PACS, each of the four domains in PACS is used as the source domain in turn, with the remaining three domains serving as target domains. This process is repeated for each domain as the source, and the model performance on the target domains is averaged to obtain the results in the table. For CIFAR-10, the training domain is CIFAR-10 itself, and CIFAR-10-C is the target domain.
>
> **[Multiple Runs]** Yes, we've run the experiments three times and provided the average results.
>
> **[Hyper-parameter choosing]** Following previous works[43,36,6,3,46], there is an external validation set for choosing models to evaluate the performance on the unseen target domains. We choose the hyper-parameters with a hyper-parameter study on the model generalization performances.
>
> **[In-domain performances]** In the SDG scenario, we generally don't focus much on in-domain performances[37,38,22,43]. We follow your suggestion to provide in-domain performance comparisons on PACS. As shown in Tab. Re4, compared with prior related SOTAs, we generally outperform them. Despite our method only achieving similar in-domain performance against ABA[3] and ALT [6] on the “Photo” source domain, we outperform the related SOTAs across all the metrics with clear advantages.
>
> **[Explanation of Fig. 3(a)]** We are sorry for the confusion caused. Fig 3(a) should be referred to in Sec. 5.4.1. In Sec. 5.4.1 we would like to justify that, given the different frequency preferences of the baseline and ours as shown between Fig. 3(a) and (b), the proposed techniques could indeed modify the model’s frequency preferences. We would further revise the manuscript and provide an illustration of Fig. 3(a) in Sec. 5.4.1.
>
> **[Baseline tSNE plots]**
>  We follow your suggestion to provide a baseline tSNE visualization in Fig. Re1, with ERM baseline and AugMix baseline. As can be seen, feature distributions in the baselines are more distinct from each other, indicating a weaker signal of generalization performance compared with our method.
>
> ## Module design choices in AAUA.
>
> **(1) [Intuition]** Intuitively, low-frequency components are the dominant ones in real-world datasets, which could have a higher possibility of containing frequency shortcuts. Therefore, we apply the current design of forcing AAUA to inject aggressive adversarial perturbation into low-frequency components only.
>
> **(2) [Justification]** Following your suggestion, we further provide a comparison with different settings of the AAUA in terms of generalization performances. As shown in Tab. Re5, we compare the low-frequency-centered settings (Ours) with two settings from your advice. One is applying it to both low and high frequency components(All), and only on the high frequency components (Reverse). As can be seen, our setting achieves the strongest performances, with advantages over the “All” setting, and significantly outperforms the “Reverse” setting. The experiment results justify the low-frequency-centered design of AAUA.
>
> We sincerely thank you for this valuable comment, and we will further clarify it in the revised manuscript with the intuitive motivation and ablation study justification of the AAUA design.

---

> ### Comment · Reviewer_jMvT · 2024-08-12
> **Official Comment**
>
> I thank the authors for their detailed response.
> * It is nice to see that the proposed method can get better domain generalization without sacrificing ID accuracy on PACS.
> * The ablation study in table Re5 is interesting. It suggests that some domains (P,C) need AAUA on low frequencies for good performance, while others don't.
>
> However, I still have some concerns
>
> * I am still not clear on the hyper-parameter selection. Looking at [43,46], I was unable to figure out how the hyper-parameters were selected in these works either. This is a crucial point, since it is easy to over-state the empirical results of a method by "leaking" test domains through hyper-parameters, e.g. see Gulrajani and Lopez-Paz, 2020. I would urge the authors to detail the exact mechanism behind their hyper-parameter selection method, i.e. define the space of hyperparameters searched, and specify the "external validation set" used.
> * With regards to multiple run, I would like to know what the standard deviations were, and if the reported improvements in performance are significant. I stress on this point because the absolute value of improvements is low over previous methods.
>
>
>
>
> Gulrajani, Ishaan, and David Lopez-Paz. "In search of lost domain generalization." arXiv preprint arXiv:2007.01434 (2020).

---

> > ### Author Response · Authors · 2024-08-13
> > **Discussion Post 1**
> >
> > Thank you for taking time to engage in the discussion.
> >
> > 1. As Gulrajani and Lopez-Paz, 2020 (DomainBed) discuss multi-source domain generalization, they are slightly different from the setting of single-source domain generalization. In the single-source domain generalization paradigm, the sole source domain is partitioned into training and validation sets, with an 8:2 ratio [3，6]. The model is trained using the training set and subsequently evaluated on the validation set. The model iteration that demonstrates optimal in-domain performance on the validation set is then selected for testing/ evaluation on the entirety of the target domain data.
> >    Regarding the hyper-parameter selection, hyper-parameter studies are provided on the method-related hyper-parameters in Fig. 8 of appendix and Tab. Re6.Re7 in the rebuttal PDF. We didn't conduct hyper-parameter search on the other parameters (i.e. learning rate and batch size), but instead pre-defined them. This is aligned with the protocol in prior works[3,6]. These pre-defined values are provided in Sec. A.4 in the appendix.
> >
> > 2. As most of the single-source DG works in Tab. 1 didn't provide standard deviation in their paper, we thus previously excluded this term. We follow your suggestion to provide the performance along with the standard deviations below:
> >
> > | Method | Digits       | PACS         |
> > | ------ | ------------ | ------------ |
> > | ABA    | 74.76 (0.52) | 66.02(1.15)  |
> > | ALT    | 72.49 (0.87) | 64.72 (-)    |
> > | Ours   | 81.39 (0.58) | 67.91 (0.57) |

---

> > > ### Comment · Reviewer_jMvT · 2024-08-13
> > > **Official comment**
> > >
> > > I thank the authors for their response.
> > >
> > > * The protocol for hyper-parameter selection is now clear to me. I missed the discussion in Appendix A.7 in my original review, and I apologise for the over-sight. One good thing is that the authors do not tune per-domain hyper-parameters, which could lead to an unfair comparison.
> > > * However, the hyper-parameter study is a bit concerning to me in terms of the results. It appears that the bulk of the accuracy improvement of the method on PACS come from the Photo domain. However, across all hyper-parameter sweeps, the Photo domain performance is especially sensitive to the correct hyper-parameters. This might cast some doubt on the ease of use of the method, and should be stated upfront by the authors while discussing the results.
> > > * The standard deviations provide more information about the method's empirical performance, and it seems like the method marginally out-performs ABA on PACS. The proposed method's lower variance is a positive trait. However, the claim of out-performing the state of the art by large margins needs to be toned down in the writing of section 5. I would also encourage the authors to include this information in future submissions as well, especially for methods whose performance is close to the proposed method (e.g. [36,43] on CIFAR-10C to further contextualize the empirical performance of the method.
> > >
> > >
> > > Overall, I believe that the paper needs to account for these concerns to make the claims of better empirical performance more water-tight. I am raising my score to marginally above accept, but **I believe that the presentation can be improved upon greatly**, and would encourage the authors to keep that in mind for future submissions.

---

> > > > ### Author Response · Authors · 2024-08-14
> > > > **Official Comment**
> > > >
> > > > Thank you for taking the time to review our submission and for raising the score. We truly appreciate your constructive feedback and insights.
> > > > We are committed to implementing all the suggested changes in the final version to enhance the quality of our work.
> > > >
> > > > Thank you once again for your support.

---

### Official Review · Reviewer_uwHe · 2024-07-30

**Soundness:** 2
**Presentation:** 3
**Contribution:** 3
**Rating:** 6
**Confidence:** 4

**Summary:**

This paper aims to develop data augmentation techniques to prevent the learning of frequency shortcuts and achieve enhanced generalization performances. They modify the frequency spectrum of the dataset statistical structure with aggressive frequency data augmentation, aiming to adaptively manipulate the model’s learning behavior on various frequency components. This paper proposes two effective and practical adversarial frequency augmentation modules, AAUA and AAD, designed to alter the frequency characteristic and prevent the learning of frequency shortcuts.

**Strengths:**

1. This paper explains and analyzes the frequency bias in neural network learning.
2. The authors propose two effective and practical adversarial frequency enhancement modules, which are designed to dynamically alter the frequency characteristics of the dataset.
3. The paper combines data augmentation and frequency analysis to address the learning behavior of frequency bias.

**Weaknesses:**

1.According to Figure 2, you used the AAUA and AAD networks for data augmentation. We would like to know the parameters and time complexity of the AAUA and AAD networks. Has the increase in parameters led to improved effectiveness? Do we still need to train your networks separately? How does the training time compare to that of the main network? Can we directly use the pre-trained parameters you provide for most networks?
2. There are currently many research methods such as splicing, augmentation, mixup, cutmix, Mosaic, etc. How is their performance in terms of frequency? What is the difference between them and yours?
3. The paper [1] elucidates that certain components of frequency are not conducive to network generalization, and therefore adaptive filtering out some frequencies may have additional effects if the remaining frequencies are enhanced. What are the differences between your paper and [1]? Can you compare it with [1]?
[1] Deep frequency filtering for domain generalization[C]//Proceedings of the IEEE/CVF conference on computer vision and pattern recognition. 2023: 11797-11807.

**Questions:**

Please see the Weaknesses.

**Limitations:**

Please see the Weaknesses.

---

> ### Author Rebuttal · Authors · 2024-08-06
>
> ## Parameters and Time complexity.
>
> **(1) [No Additional Parameters]** We sincerely thank you for your insightful comments. Notably, as AAUA and AAD are data augmentation methods instead of neural networks, they don’t have any additional parameters. Thus, there won’t be additional parameters to train.
>
> **(2) [Additional Training Time and Efficency Study]**
> There would be additional training time costs for generating the augmentation samples. We provide an efficiency study in terms of running time per epoch and performance improvement over baseline (ERM), including our method and related adversarial augmentation works[3,6,46] on PACS. Experiment results are shown in Tab. Re1. The time is calculated on a single NVIDIA A100-80G, with the same batch size of 32. The "Photo" domain of PACS is used as the training domain.
>
> As can be seen, compared with the related adversarial augmentation SOTAs[3,6,46], we achieve the best performances with minimal additional training time over ERM, demonstrating the superior efficiency of our method. We reckon this is attributed to the parameter-free characteristic (i.e. ABA has an additional Bayesian network to help generate augmentation) and smaller iterative optimization steps of our method (Ours:5 steps, AdvST[46]: 50 steps, ALT[6]:10 steps, ABA[3]:14 steps).
>
> **(3) [Simple Application]**
> To apply AAUA and AAD, one would only need to set the hyperparameters for them and input the images with corresponding labels.
>
> ## Frequency Shortcut Evaluation with Traditional Augmentation methods.
>
> **(1) [Additional Experiments on Frequency Shortcut Evaluation]**
> We follow your suggestion to conduct frequency shortcut evaluation on more augmentation methods, as shown in Tab. Re2. The experiment results demonstrate the effectiveness of our method in preventing frequency shortcut learning, while the traditional augmentation techniques fail to do so and leads to hallucinations of generalization ability improvement by applying more frequency shortcuts.
>
> **(2) [Method Differences]**
> The core difference between our method and the recommended ones is that we conduct augmentation in the Fourier domain, while the mentioned methods focus on the spatial domain. Further, adversarial learning is also applied to enhance the augmentation samples' learning difficulty and to adaptively disrupt the learned frequency shortcuts.
>
> ## Difference with Deep Frequency Filtering (CVPR'23, not open-source)
>
> **(1) [Data-centered vs Network-centered]**
> Ours is a data-centered method, and DFF is a network-centered method. We focus on preventing frequency shortcut learning with aggressive frequency augmentation, while DFF aims to learn adaptive Fourier filters for intermediate features.
>
> **(2) [Experiment Comparisons]**
> Following your suggestion, we provide a comparison with DFF (implemented by us, as the code of DFF is not open-source) in terms of frequency shortcut evaluation and performances on single-domain generalization in Tab. Re3.
> As DFF learns the filter adaptively, with no additional regularization, it eventually learns more frequency shortcuts. Further, experiment results indicate that DFF would fail in the single-source domain generalization scenario.

---

### Author Rebuttal · Authors · 2024-08-07

We thank the reviewers (uwHe, jMvT, 57HS, szDc) for all the informative and constructive feedback and we appreciate the comments to improve our work.

**Reviewer uwHe**: "Two effective and practical adversarial frequency enhancement modules. Combines data augmentation and frequency analysis to address the learning behavior of frequency bias."

**Reviewer jMvT**: "Extensive ablation studies. The method is benchmarked on a variety of classification and retrieval tasks, and superior performance is demonstrated."

**Reviewer 57HS**: "Innovative data augmentation techniques, provide a new perspective. The theoretical foundation is robust and experimental validation is thorough."

**Reviewer szDc**: "Well-written introduction. Interesting and insightful ablation study."

We've provided detailed explanations to address each concern from the reviewers with additional experiment results attached in the individual PDF file. We summarize the main points presented in our response and we kindly hope that we have addressed all the concerns.

- We address the questions and provide clarifications and details that have improved our work.
- We include comparisons with more papers and experiments on large-scale datasets to show the effectiveness of our method.
- We include more variants of our proposed method that bring further insight into our choice.
- We will update the final version to incorporate all the additional results with the reviewers' comments.

---

### Decision · Program_Chairs · 2024-09-25

**Decision:**

Accept (poster)

**Comment:**

The paper received mixed ratings at the first place. After discussion, Review szDc, who gave the most negative rating, changed his/her mind to increase the score to 5. leading to all positive, borderline accept ratings. After carefully reading the paper and all the rebuttal, the decision is a borderline acceptance. While the average score is not high enough, the paper truly has some merits, including novel problem setting, sufficient theoretical analysis, and compeling experimental results. The following is a summary of the review.

Reviewers acknowledge the innovative approach of using adversarial frequency augmentation techniques, AAUA and AAD, to mitigate frequency shortcuts in neural networks and improve domain generalization (reviewer uwHe, 57HS). However, the paper is criticized for lacking detailed explanations of the computational complexity, hyperparameter sensitivity, and training costs associated with the proposed methods (reviewer uwHe). Additionally, the experimental design and presentation are seen as problematic; the clarity of implementation details is insufficient, with arbitrary design choices and unclear motivations for some decisions (reviewer jMvT). The evaluations are limited in scope, involving only small datasets and missing comparisons with relevant frequency augmentation techniques, which undermines the generalizability and applicability of the findings (reviewer szDc). Moreover, inconsistencies in the theoretical justification and an unclear distinction from prior work further weaken the contribution (reviewer szDc). The writing and structure also need improvement, as the presentation is difficult to follow and lacks coherence in explaining the experimental results (reviewer jMvT).

Overall, while the idea of addressing frequency bias in domain generalization is recognized as valuable.